# Structural and biochemical characterization of a novel thermophilic Coh01147 protease

Hossein Tarrahimofrad[1,2], Amir Meimandipour[1], Sareh Arjmand[3], Mohammadtaghi Beigi Nassiri[2], Ehsan Jahangirian[1], Hossein Tavana[4], Javad Zamani[1], Somayyeh Rahimnahal[1,2], Saeed Aminzadeh[1] *

1 Bioprocess Engineering Group, Institute of Industrial and Environmental Biotechnology, National Institute of Genetic Engineering and Biotechnology (NIGEB), Tehran, Iran, 2 Department of Animal Science and Food Technology, Agriculture Science and Natural Resources University Khouzestan, Ahwaz, Iran, 3 Protein Research Center, Shahid Beheshti University, G. C., Tehran, Iran, 4 Department of Biomedical Engineering, The University of Akron, Akron, OH, United States of America

* aminzade@nigeb.ac.ir

**Data Availability Statement:** All relevant data are within the paper and its Supporting Information files.

## Abstract

Proteases play an essential role in living organisms and represent one of the largest groups of industrial enzymes. The aim of this work was recombinant production and characterization of a newly identified thermostable protease 1147 from thermophilum indigenous Cohnella sp. A01. Phylogenetic tree analysis showed that protease 1147 is closely related to the cysteine proteases from DJ-1/ThiJ/PfpI superfamily, with the conserved catalytic tetrad. Structural prediction using MODELLER 9v7 indicated that protease 1147 has an overall α/β sandwich tertiary structure. The gene of protease 1147 was cloned and expressed in Escherichia coli (E. coli) BL21. The recombinant protease 1147 appeared as a homogenous band of 18 kDa in SDS-PAGE, which was verified by western blot and zymography. The recombinant protein was purified with a yield of approximately 88% in a single step using Ni-NTA affinity chromatography. Furthermore, a rapid one-step thermal shock procedure was successfully implemented to purify the protein with a yield of 73%. Using casein as the substrate, $K_m$, and $k_{cat}$, $k_{cat}/K_m$ values of 13.72 mM, 3.143 × 10$^{-3}$ (s$^{-1}$), and 0.381 (M$^{-1}$ S$^{-1}$) were obtained, respectively. The maximum protease activity was detected at pH = 7 and 60˚C with the inactivation rate constant (kin) of 2.10 × 10–3 (m$^{-1}$), and half-life (t$_{1/2}$) of 330.07 min. Protease 1147 exhibited excellent stability to organic solvent, metal ions, and 1% SDS. The protease activity was significantly enhanced by Tween 20 and Tween 80 and suppressed by cysteine protease specific inhibitors. Docking results and molecular dynamics (MD) simulation revealed that Tween 20 interacted with protease 1147 via hydrogen bonds and made the structure more stable. CD and fluorescence spectra indicated structural changes taking place at 100˚C, very basic and acidic pH, and in the presence of Tween 20. These properties make this newly characterized protease a potential candidate for various biotechnological applications.

**Funding:** Yes Saeed Aminzadeh, National Institute of Genetic Engineering and Biotechnology (NIGEB) grant number 971215-I-712

**Competing interests:** The authors have declared that no competing interests exist.

## Introduction

Proteases represent a well-defined class of hydrolytic enzymes that are broadly used in numerous manufacturing processes such as detergents, leather, food, feed, and waste treatment industries. They represent one of the most important classes of commercial enzymes that account for almost a quarter of the total enzyme manufacturing worldwide [1]. Proteases act as "sharp scissors" that catalyse specific or non-specific digestive processes to produce new protein products, small peptides, or even individual amino acids. These enzymes play a vital role in a variety of biological processes, including cell growth, death, and immune responses [2].

Protease enzymes can be isolated from eukaryotic and prokaryotic sources. However, microorganisms represent the main source for the production of proteases due to their various economic and technical advantages. The culture of protease-producing microorganisms in high quantities and in a short time can be done using established fermentation methods. Furthermore, microorganisms facilitate the production of proteases with broad natural biochemical diversity and engineering of novel enzymes through genetic manipulations [3,4].

A major impediment to the commercial use of enzymes is the thermal stability constraint required for many industrial practices such as in the dairy industry and laundry detergent manufacturing [5–7]. Although several heat-stable proteases have been isolated from the thermophilus bacteria or produced through genetic engineering [8–10], there is still a strong need to identify and characterize new thermostable proteases from different microorganisms for environmental and industrial applications.

The goal of the present study was the recombinant expression of a previously isolated thermostable protease gene (Coh01144) from the thermophilic indigenous *Cohnella sp*. A01 [11]. The recombinant protease was purified and characterized in terms of biophysical and biochemical properties such as molecular docking and molecular dynamics simulation, thermal stability, pH profiling, kinetic parameters, and thermodynamic analysis to identify the potential industrial application of this protease.

## Materials and methods

### Strains, plasmids, reagents, and chemicals

*E. coli* strains DH5α and BL21 (DE3), and pET26b (+) vector were purchased from Invitrogen (USA). Restriction enzymes were provided by Fermentas (USA). PCR product extraction and high pure plasmid purification kits were purchased from Bioneer (Korea). High pure DNA purification kit was purchased from Roche (Germany). T4 DNA Ligase, isopropyl β-D-1-thio-galactopyranoside, and PVDF membranes were purchased from Thermo Fisher Scientific (Germany). Casein, azo-casein, 1 anilino naphthalene-8-sulfonate (ANS), and anti-polyHistidine peroxidase antibody were purchased from Sigma (USA). All other chemicals were obtained from Merck (Germany).

### Structural studies

***In silico* structural and phylogenetic analysis.** The nucleotide sequence of Coh01147, previously obtained from *Cohnella sp*. A01 (Genbank Accession No: MK645908) was translated to the amino acids using Expasy translate tool (http://web.expasy.org/translate). The resulting protein sequence was used for further *in silico* analysis of the *Cohnella sp*. A01 derived protease 1147 protein. The biochemical properties of the protein, including molecular weight, isoelectric point, instability index, aliphatic coefficient, and grand average of hydropathy (GRAVY), were calculated using Expasy ProtParam tool (http://web.expasy.org/protparam) [12]. The presence of putative signal peptide and the location of its cleavage site

was predicted using SignalP 5.0 server (http://www.cbs.dtu.dk/services/SignalP). pFam (http://pfam.xfam.org/search) and MEROPS (http://MEROPS.sanger.ac.uk) databases were used for sequence alignment and identifying the peptidase family. The phylogenetic relationships of the protein sequence were analysed using ClustalW in MEGA 7 programs. Multiple alignments of the resulting protein family of protease 1147 (from MEROPS) was carried out using COBALT (https://www.ncbi.nlm.nih.gov/tools/cobalt/re_cobalt.cgi), and the conserved areas were determined using DNAMAN software (www.lynnon.com). BLOSUM62 substitution matrix and a gap penalty = 12 were utilised in the search [13].

**Homology modeling and validation.** BLASTP (at an E-value cutoff of $< 3.5E^{-106}$ with BLOSUM62 matrix) was used for searching homologous sequences of protease 1147 in the protein data bank (PDB). The query resembled the 3D crystal structure of an intracellular protease from *Pyrococcus horikoshii* (PH1704, PDB ID: 1G2I_A) at a 2-Å resolution that was used as a template to construct the protease 1147 model. MODELLER 9v7 [14] was used to generate the homology model of the query. To evaluate and validate the modeled 3D structure, a Ramachandran plot was constructed using PROCHECK server (http://services.mbi.ucla.edu/PROCHECK) at the SAVE server (http://services.mbi.ucla.edu/SAVES). The ProSA web server (https://prosa.services.came.sbg.ac.at/prosa.pHp) was employed to calculate the Z-Score and evaluate the consistency between the crystal structure of the template (PH1704) and modeled protease 1147 [15]. The Chimera v1.13.1 was used to calculate the optimal energy of predicted protease 1147 structure and select the best model with the least root mean square deviation (RMSD) value by superimposing the model with the template [16, 17]. The active site of the proposed model was predicted using 3DLigandSite (http://www.sbg.bio.ic.ac.uk/3dligandsite/) and evaluated using the Pymol program (http://pymol.sourceforge.net/). The similarity of the predicted binding sites of protease 1147 and PH1704 was compared using Chimera v1.13.1.

**Protein-ligand docking studies.** The modeled protease 1147 was docked with two surfactants with a positive effect on its activity (tween 20 and SDS) using Molegro Virtual Docker V.6.0 (MVD). The predicted structure was minimized, and the potential cavity (also referred to as the active site) was identified using the built-in cavity algorithm of MVD. The chemical structures of ligands were obtained from the PubChem and stored as Mol2 files using the Chimera v1.13.1. Finally, the structure of the protein was docked against the ligands using the AuotDock Vina tool. Grid size for the protein-ligand complex was set to $12 \times 10 \times 14$ points. For each of the protease-SDS and protease-tween 20 complexes, 10 test runs were performed. Docked complex with lower binding energy was selected for MD simulation. Docking computations were performed in triplicate.

**Molecular dynamics simulation.** The prepared protein and protein-ligand complexes (protease-SDS and protease-tween complexes) were used in MD simulation, performed using GROMACS v 4.6.5 with CHARMM36 all-atom force field. The CGenFF server (https://cgenff.umaryland.edu/) provides topologies and parameters of ligands compatible with the CHARMM36 all-atoms force field. Protein and protein-ligand complexes were soaked in a cubic box of water molecules, and the charges on the protein were neutralized by the addition of $Na^+$ and $Cl^-$ ions. The energy of the system was minimised using the steepest descent algorithm to eliminate bad contact and clashes. The NVT and NPT ensembles were used during the equilibration to achieve the desired temperature (373.15 K for protease and 300 K for protease-ligand complex) and pressure (1 bar) for 100 ps and restraint forces of 1000 kJ/mol. Finally, 40 ns MD run were performed in triplicate after releasing all restraints. All bonds were constrained by the LINCS algorithm [18]. dx.doi.org/10.17504/protocols.io.bgcajsse

## Cell culture and DNA extraction

The *Cohnella sp*. A01 cells were primarily cultured in a nutrient broth medium (0.5% peptone, 0.3% yeast extract, 0.5% sodium chloride, soluble in distilled water, pH adjusted to 6.8) overnight at 60˚C on a shaker incubator (180 rpm). Of the culture medium, 2 ml ($3.5 \times 10^8$ cells/ml) was added to 50 ml of the fresh nutrient broth medium and incubated in the same condition for 5 more days. The harvested cells were centrifuged at $3500 \times g$, and 4˚C for 30 min and their genomic DNA was isolated using a high pure DNA purification kit, according to the manufacturer's instructions.

## Cloning and heterologous expression of protease 1147 gene

Protease 1147 gene was amplified by PCR using specific primers (forward `5'-TACATAT GAAGAAAGTCGCTTTCCTGC-3'` and reverse `5'-TACTCGAGGCTCAGCTTGTTCAGCGT TTC-3'`) carrying *Nde*I and *Xho*I restriction enzyme recognition sites. The reverse primer was designed without stop codon to incorporate the His-tag sequence at the C-terminal of the protease. The purified PCR product was digested with the aforementioned restriction enzymes and cloned into the expression vector pET26b(+) using the T4 ligase. The recombinant plasmid pET26b(+) was transformed into *E. coli* DH5α through the heat shock transformation method, and the verified recombinant plasmid pET26b(+) was transformed into *E. coli* BL21 (DE3). A positive BL21 bacterial colony was incubated overnight at 37˚C in 5 ml LB medium containing kanamycin (30 mg/ml) and inoculated to the fresh medium with the same antibiotic concentration. After the $OD_{600}$ reached 0.6, the expression of the recombinant protein was initiated by the addition of 1 mM IPTG and incubating overnight at 28˚C (150 rpm).

## Recombinant protein purification

**Ni-NTA affinity chromatography.** After the cultivation period, cells were collected by centrifugation at $3500 \times g$ for 30 min at 4˚C and resuspended in 4 ml lysis buffer (50 mM $NaH_2PO_4$, 300 mM NaCl, 10 mM Imidazole and 0.05% Tween 20 at a pH of 7). The mixture was sonicated with $4 \times 40$ sec pulses followed by 20 sec rest between cycles at 4˚C. The crude extracts were then centrifuged at $9000 \times g$ for 30 min at 4˚C, and the resulting supernatant was subjected to Ni-NTA purification according to a previously described protocol [8]. In brief, 4 ml of sample was resuspended in binding buffer and loaded onto the Ni-NTA resin pre-equilibrated in 6 ml equilibration buffer (50 mM $NaH_2PO_4$, 300 mM NaCl, 10 mM Imidazole and 0.05% Tween 20% at pH = 7). Then, the column was washed three times with a buffer (50 mM $NaH_2PO_4$, 300 mM NaCl, 500 mM Imidazole and 0.05% Tween 20 at pH = 7) and four times with an elution buffer (50 mM $NaH_2PO_4$, 300 mM NaCl, 1200 mM Imidazole and 0.05% Tween 20 at pH = 7). The purified elution fractions were dialysed overnight in a 50 mM potassium-phosphate buffer (pH = 7) at 4˚C. All the collected fractions were visualised using 12% sodium dodecyl sulfate-polyacrylamide gel electrophoresis. Protein bands were stained by Coomassie brilliant blue R250. Soluble purified protein concentration was measured using the Bradford method as described previously [19].

**Single-step purification by heat shock.** Heat treatment was used as the second method for the protease 1147 purification. For this purpose, the supernatant sample obtained in the previous step was heat-treated in a hot water bath at 90˚C for 15 min. Insoluble material was separated by centrifugation at $13000 \times g$ for 10 min, and the supernatant, containing the enzyme, was analyzed on a 12% SDS–polyacrylamide. dx.doi.org/10.17504/protocols.io. bgbsjsne

## Western blotting

After separation on the 12% SDS-PAGE, the proteins were transferred onto a PVDF membrane for Western blotting. The transferred membrane was blocked at room temperature for 5 h with TBST (25 mM Tris–HCl, pH 7.4, 0.14 mM NaCl, and 0.05% Tween 20) containing 5% BSA. The membrane was washed three times with TBST and incubated for 3 h in a 1:2000 diluted monoclonal anti-polyHistidin peroxidase at 37˚C. After three more washes with the TBST, the target protein was visualized by developing the blot for 30 min with 3.4 mM 4-chloro-1-naphthol and 0.04% (v/v) $H_2O_2$ as a substrate.

## Protease activity assay and kinetic measurements

To investigate the substrate specificity for the protease 1147, the enzyme activity was measured using a variety of commercially available protease substrates, including casein, BSA, gelatin, and azo-casein (1% w/w). Measurements of protease activity were performed according to Gulmez et al. [8] and with slight modification. In brief, to prepare the substrate solution, 250 μl of the substrate was dissolved in a 50 mM potassium phosphate buffer. The solution was mixed with 50 μl of protease 1147 enzyme and incubated at 60˚C for 30 min. Of a fresh and cold 10% Trichloroacetic acid (TCA) solution, 300 μl was used to stop the reaction. After 1 h of incubation on ice, the reaction mixture was centrifuged for 10 min at 10000 × g and 4˚C. The absorbance of the supernatant was measured at 280 nm.

Kinetic parameters of the purified protease 1147 were characterized in terms of Michaelis/Menten kinetic parameters ($K_m$, $V_{max}$, $k_{cat}$, and $k_{cat}/K_m$) using non-linear regression by GraphPad Prism V.8 [20]. The protease activity was determined in the presence of a different concentration of casein (ranging from 10 to 70 mM) as substrate and at the optimum temperature. Assays were performed with 0.7 mg/ml of the purified enzyme and in triplicate [21].

## Zymography

Casein Zymography was performed on SDS-PAGE according to the method of Garcia-Carreno et al. [22]. After electrophoresis, the gel was shaken for 1 h in 50 mM Tris-HCl buffer containing 2.5% Triton X-100 at 4˚C (pH = 7) to remove SDS. To remove the Triton X-100, the gel was washed three times with 100 mM Tris–HCl buffer (pH 7). The gel was incubated for 1 h at 60˚C in potassium phosphate buffer 50 mM, contained casein 1%, and subsequently stained with Coomassie Brilliant Blue R-250. Clear zones on the blue background indicated the presence of protease activity.

## Effects of temperature and pH on the protease activity and stability

The effect of temperature on the protease activity was evaluated by measuring the enzyme activity at different temperatures from 10 to 100˚C and with 10˚C intervals. The 1% (w/w) casein was used as the substrate. To explore the temperature stability of the protein, the protease activity was measured at 60, 70, 80, and 90˚C for 2 h and with 10 min intervals. All assays were performed in triplicate.

To determine the pH profile of the enzyme, the protease activity was performed by casein substrate in a pH range of 3 to 11 using 50 mM acetate (pH 3.6–5.6), 50 mM phosphate (pH 5.8–8.0), and 50 mM glycine (pH 8.6–10.6) buffers. An optimum temperature (60˚C) was used for the assays. To investigate the pH stability, the residual activity was measured at pH values of 5, 9, and 11 for 2.5 h at 60˚C and with 10min intervals.

## Effects of metal ions, protease inhibitors, detergents, organic solvents on proteolytic activity

The effect of metal ions on protease activity was determined by measuring the enzymatic activity of the purified enzyme in the presence of 1, 2, and 5 mM concentrations of various metal ions including $Fe^{2+}$, $Mn^{2+}$, $Mg^{2+}$, $Ca^{2+}$, $Co^{2+}$, $Zn^{2+}$, $Na^+$, $K^+$, $Al^{3+}$, and $Li^+$. The effect of organic solvents was determined using the 5 and 10% (v/v) concentrations of acetone, methanol, ethanol, isopropanol, isobutanol, glycerol, DMSO, n-hexane, and chloroform as organic solvents in the reaction of enzyme activity measurements. The effect of surfactants on protease activity was evaluated using 1 and 2% concentrations of four different surfactants, i.e., Tween 20, Tween 80, Triton 100X, and SDS. To evaluate the effect of protease inhibitors, the enzyme activity was assayed in the presence of 1 and 2% concentrations of iodoacetamide (IAA), guanidinium hydrochloride (GuHCl), dithiothreitol (DTT), phenylmethylsulfonyl fluoride (PMSF), E-64 [Trans-Epoxysuccinyl-L-leucylamido (4-guanidino) butane], Leupeptin, and ethylenediaminetetraacetic acid (EDTA). The activity of the enzyme solution containing no metal ion was set as 100%, and the residual protease activity was measured.

## Thermodynamic study

As described by Papamichael et al. the enzyme activation energy, and irreversible thermal inactivation can be described as the basic relations associating the changes of Gibbs free energy ($\Delta G$), enthalpy ($\Delta H$), and entropy ($\Delta S$) to the values of equilibrium constants through absolute temperature and/or temperature stability [23].

The changes in the enthalpy and entropy of activation energy ($\Delta H^{\ddagger}$, $\Delta S^{\ddagger}$) to form enzyme/substrate complex [ES] were calculated from the rearranged Eyring absolute rate equation [23]:

$$k_{cat} = (k_B/\hbar) \times T \times e^{-\Delta G\ddagger/RT} = (k_B/\hbar) \times T \times e^{-\Delta H\ddagger/RT} \times e^{\Delta S\ddagger/R} \tag{1}$$

where, $k_B$ is the Boltzmann constant (i.e. R/N, which is $1.38 \times 10^{-23}$ J/K), N is the Avogadro's number ($6.02 \times 10^{-23}$ mol$^{-1)}$, T is the temperature in Kelvin, $\hbar$ is the Planck constant ($6.63 \times 10^{-34}$), R is the gas constant (8.314 J/K mol), $\Delta S^{\ddagger}$ is the change in the entropy, and $\Delta H^{\ddagger}$ is the change in the enthalpy. The $\Delta H^{\ddagger}$ and $\Delta S^{\ddagger}$ values were calculated from the slope and intercept, respectively, of Ln[$k_a$/T] vs. 1/T as follows:

$$k_{cat} = [(k_B/T)/\hbar] \times k_{cat}{}^{\ddagger} \tag{2}$$

$$\Delta H^{\ddagger} = E_a{}^{\ddagger} - RT \tag{3}$$

The Gibbs free energy of activation energy ($\Delta G^{\ddagger}$) of the protease was calculated from:

$$\Delta G^{\ddagger} = -RT[Ln\ k_{cat}{}^{\ddagger}] \tag{4}$$

or from the well-known equation:

$$\Delta G^{\ddagger} = \Delta H^{\ddagger} - T\Delta S^{\ddagger} \tag{5}$$

$$\Delta S^{\ddagger} = \Delta H^{\ddagger} - \Delta G^{\ddagger}/T \tag{6}$$

The energy of activation ($E_a{}^{\ddagger}$) was calculated using the Arrhenius equation:

$$k_a = Ae(-E_a{}^{\ddagger}/R) \tag{7}$$

such that

$$Ln[k_a] = -E_a^{\ddagger}/R \tag{8}$$

The energy $(E_a^{\ddagger})$ involved in the activation process was calculated from the slope of a linear plot of $1/T$ vs. Ln $[k_a]$.

Free energy of enzyme/substrate binding $[\Delta G^{\ddagger}{}_{E-S}]$ was obtained from:

$$\Delta G^{\ddagger}{}_{E-S} = -RT\ Ln\ K_i,\ where\ K_i = 1/K_m \tag{9}$$

Free energy for transition state $[\Delta G^{\ddagger}{}_{E-T}]$ formation was acquired from:

$$\Delta G^{\ddagger}{}_{E-T} = -RT\ Ln(k_{cat}/K_m) \tag{10}$$

Thermodynamic parameters of irreversible inactivation of protease 1147 were calculated using data obtained from the assay of temperature stability, as follows:

$$k_{in} = (k_B/\hbar) \times T \times e^{-\Delta G\#/RT} = (k_B/\hbar) \times T \times e^{-\Delta H\#/RT} \times e^{\Delta S\#/R} \tag{11}$$

The values of enthalpy, entropy, and Gibbs free energy of irreversible thermal inactivation $(\Delta H^{\#}, \Delta G^{\#},$ and $\Delta S^{\#})$ were calculated by applying Eqs 2, 3, and 4 with some modification including that in Eq 2 $E_a{}^{\#}(in)$ was displaced with $E_a{}^{\ddagger}$, and in Eq 3, $k_{in}$ was used instead of $k_{cat}$

The inactivation rate $(k_{in}\ m^{-1})$ was calculated by the following first-order expression:

$$k[Act]/k_{in}t = -k_{in}[Act] \tag{12}$$

which can also be expressed as:

$$Ln([Act]t/[Act]0) = -k_{in}t \tag{13}$$

Where, t is the incubation time, [Act]0 is the initial enzyme activity (i.e. enzyme activity at time 0), and [Act]t is the enzyme activity at the time 't'. The $k_{in}$ is the inactivation rate constant, calculated from the plots of Ln ([Act]t/[Act]0) vs. t.

The half-life $(t_{1/2})$ of the enzyme is defined as the time required for the enzyme to lose one half of its initial activity, and expressed as:

$$t_{1/2} = ln2/k_{in}(min)^{-1} \tag{14}$$

The energy of activation for the inactivation process $(E_a{}^{\#})$ was calculated using the Arrhenius equation:

$$k_{in} = Ae(-E_a\#/RT) \tag{15}$$

such that

$$Ln[k_{in}] = -E_a\#/RT \tag{16}$$

The energy $(E_a{}^{\#})$ involved in the deactivation process was calculated from the slope of a linear plot of $1/T$ vs. Ln $[k_{in}]$.

Decimal reduction time (D value) was defined as the time required for 90% reduction in the initial enzyme activity at a specific temperature and was calculated as:

$$D = RT/k_{in} \tag{17}$$

All the experiments were performed in triplicate, and the mean values were presented.

## Structural analysis by fluorescence spectroscopy

Intrinsic and extrinsic fluorescence measurements were performed using Cary Eclipse spectro-fluorometer (Varian Ltd., England). The measurements were made using a quartz cuvette with a 10 mm path length and at 25˚C. The excitation slit width was set at 5 nm. Changes in the intrinsic fluorescence spectra were analysed to study the effects of temperature, pH, and two surfactants (SDS and Tween 20) on the tertiary structure of protease 1147. The spectra were recorded in the range of 300–400 nm by exciting the samples at 280 nm. Purified protease with the final concentration of 400 μg/ml was dialysed against 50 mM potassium phosphate buffer and used for experiments. The protease samples were incubated for 120 min at 50–100˚C, at a pH range of 4–9 at 4˚C, and in the presence of 1% SDS and Tween 20 (1 mM) at 4˚C to study effects of temperature, pH, and the surfactants.

To study conformational changes of the recombinant protease 1147 that result in changes in its surface polarity, extrinsic fluorescent using ANS fluorescence probe was applied. The samples were excited at 375 nm, and spectra were collected from 400 to 600 nm. Effects of temperature, pH, and surfactants were assayed in the above-mentioned conditions and the presence of ANS with a final concentration of 40 μg/ml. All the experiments were performed in triplicate.

## UV-far CD analysis

Circular dichroism (CD) spectrum (190–260 nm) of 0.2 mg/ml purified 1147 was recorded in the Avive-215 spectrometer (model 215, USA (equipped with a Peltier thermostat. Proteins were dialysed against 20 mM potassium phosphate buffer (pH = 7). The effects of temperature, pH, and surfactant on the protease 1147 unfolding were analyzed in the same experimental conditions used for fluorescence spectroscopy. The results were expressed as ellipticity (θ) (mdeg.cm$^2$/dmol) using $[\theta] = (\theta \times 100 \text{ MRW})/(cl)$. The data obtained from three replicates and the percentage of secondary structures were calculated using CDNN 2.1 software. The results were deduced from the buffer CD signal and smoothed [24].

# Results

## Structural, biochemical and phylogenetic analysis of protease 1147

The DNA sequence was translated into a protein with 169 aa in length. No signal peptide was predicted by SignalP, suggesting the intracellular production of the protease 1147 (S1 Fig). Sequence alignment and phylogenetic tree analysis revealed that the protease 1147 has the highest similarity to thermostable Protease I from *Alicyclobacillus macrosporangiidus* and clade clustered with other PfpI families of intracellular proteases (Fig 1) that are characterised by their thermal stability and conserved catalytic tetrad (Glu[77], Cys[103], His[104], and Gly[105]). The conserved position of the catalytic tetrad is shown in Fig 2A. According to these results, protease 1147 is the first reported protease from *Cohnella sp*. A01 that is classified in the DJ-1/ThiJ/PfpI superfamily. Amino acids involved in the catalytic tetrad of protease 1147 are shown in Fig 2B.

The biochemical properties of protease 1147 and several related proteases were calculated by the ProtParam tool (Table A in S1 File). The predicted instability index was below 40, confirming that the protease 1147 structure is stable. The predicted aliphatic index (86.09) ascertained the high thermal stability of the protease. The negative GRAVY score denotes the hydrophilic character of the protease.

## Homology modeling, docking, and molecular dynamics

Homology models were based on the crystal structure of PH1704, which had the highest sequence similarity (47%) with the protease 1147 and lowest E-value (4E$^{-53}$). The ProSA-web

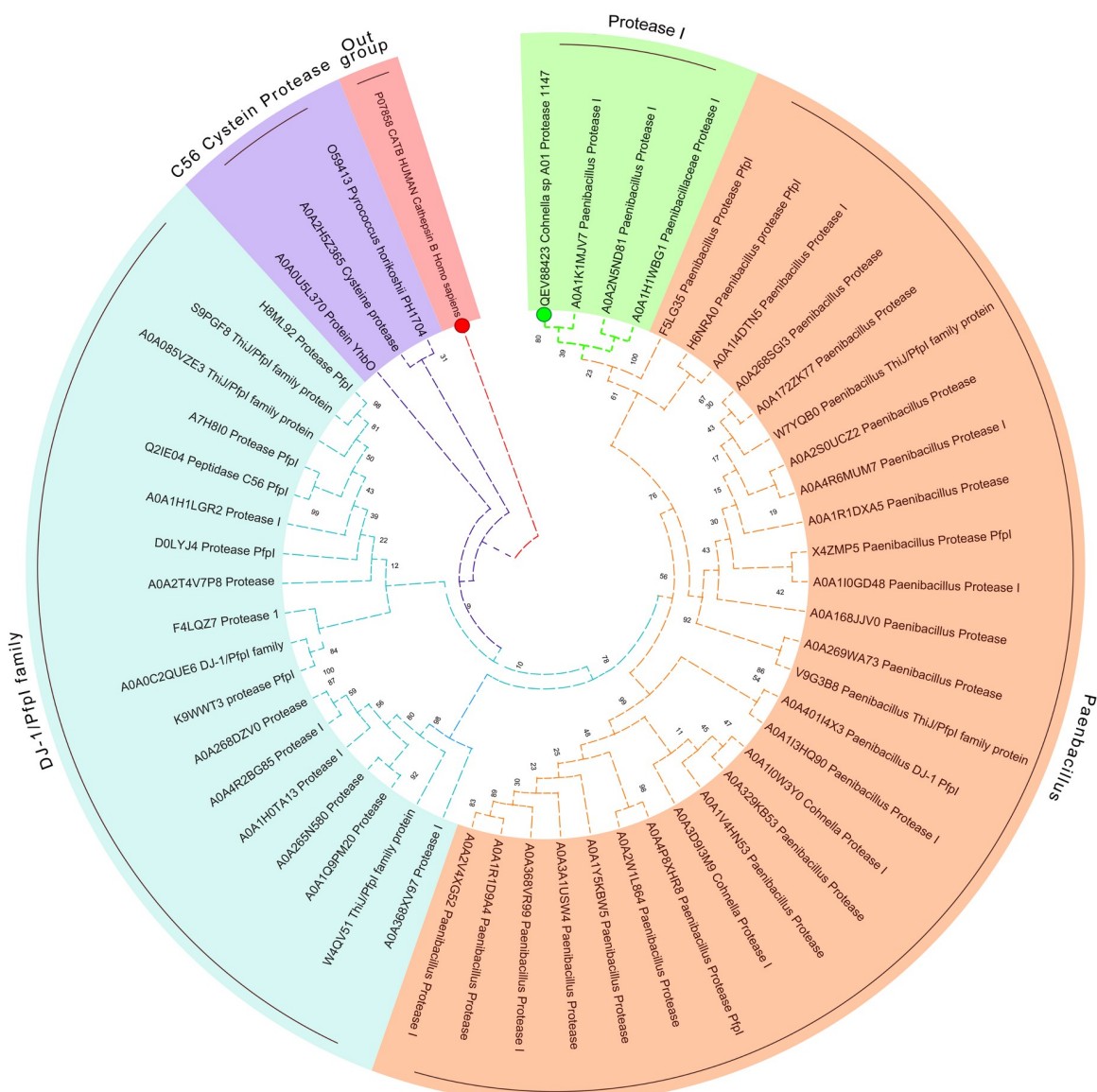

**Fig 1. Circular phylogenetic relationship of protease 1147 from *Cohnella sp*. A01 and its similar sequences.** The unrooted neighbour-joining (NJ) tree was constructed based on the alignment of the protein sequences with the high similarities. The protease 1147 is marked with a green circle. The amino acid sequence of cysteine protease human cathepsin B was used as an out-group and marked with a red circle.

has shown a Z-score of -8.34 that falls in the range of scores commonly found in the case of the similar native protein (Fig 3A). The ProSA results also confirmed that most of the residues have negative energy (S2 Fig).

The Ramachandran plot confirmed the good quality of the final model, indicating that 74%, 20%, and 4% of residues were in the favoured, allowed, and acceptable regions, respectively (Fig 3B). The superimposed structure of native *Pyrococcus horikoshii* chain A and predicted protease 1147 model is shown in Fig 3C, and also, the structural comparison of protease 1147 and PH1704 indicated that the main pocket-forming amino acids (Glu$^{77}$ –Cys$^{103}$ –His$^{104}$ –Gly$^{105}$ in protease 1147 and Cys$^{101}$ –His$^{102}$ Gly$^{103}$ in PH1704) are quite comparable. The

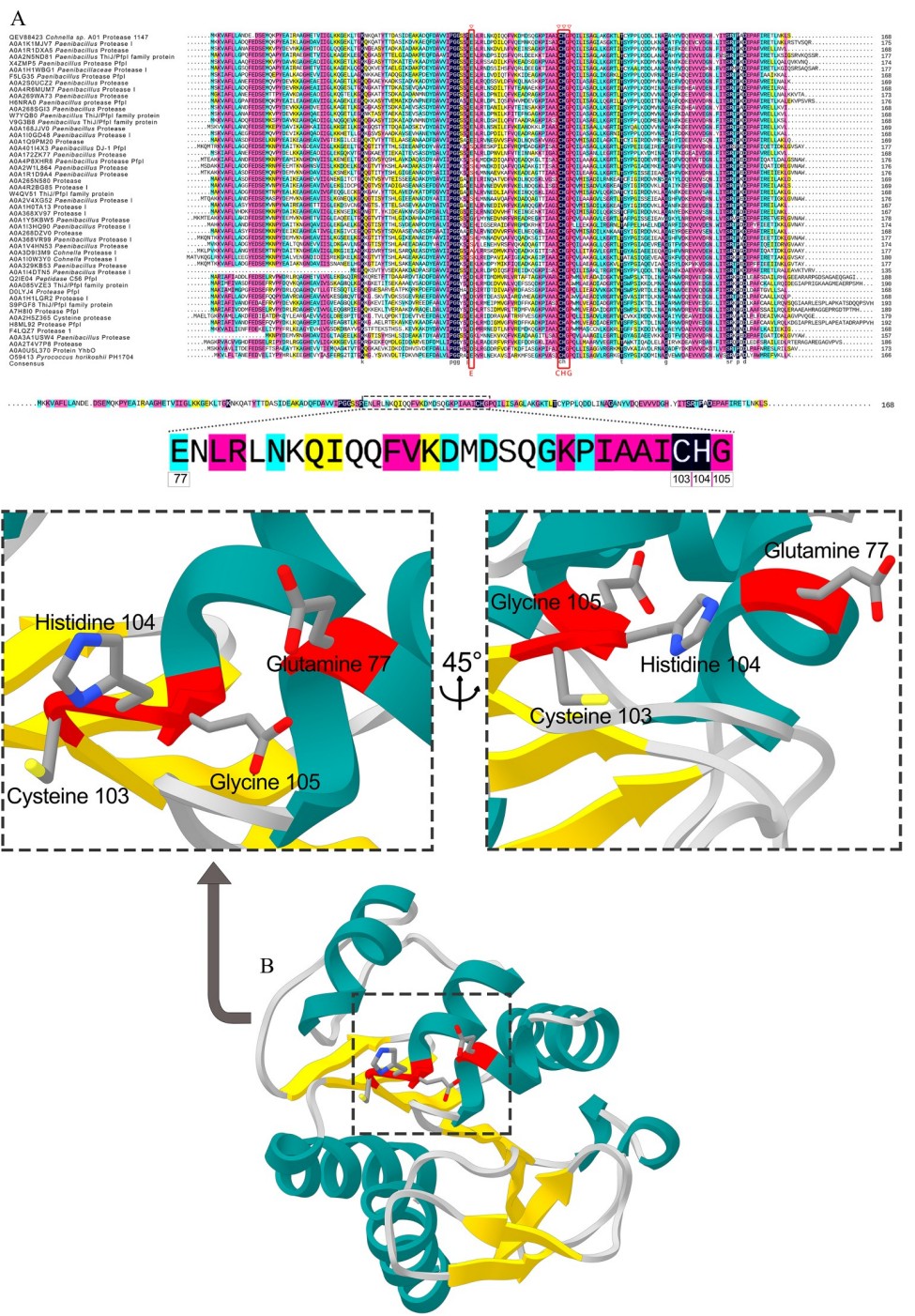

**Fig 2. Conserved sequence of protease 1447.** (A) Protein sequence alignment of the protease 1147 and several closely related sequences. The ▽ over residues in the boxed regions represent the conserved catalytic tetrad. (B) The amino acids involved in the catalytic tetrad of protease 1147 (Glu^77 –Cys^103 –His^104 –Gly^105) are marked in red ribbon sticks. The accession number of the sequences are represented.

calculated RMSD value (0.51 Å) suggests that the developed model is comparatively robust and suitable for further analysis. Catalytic tetrad residues are shown and labeled in Fig 3D.

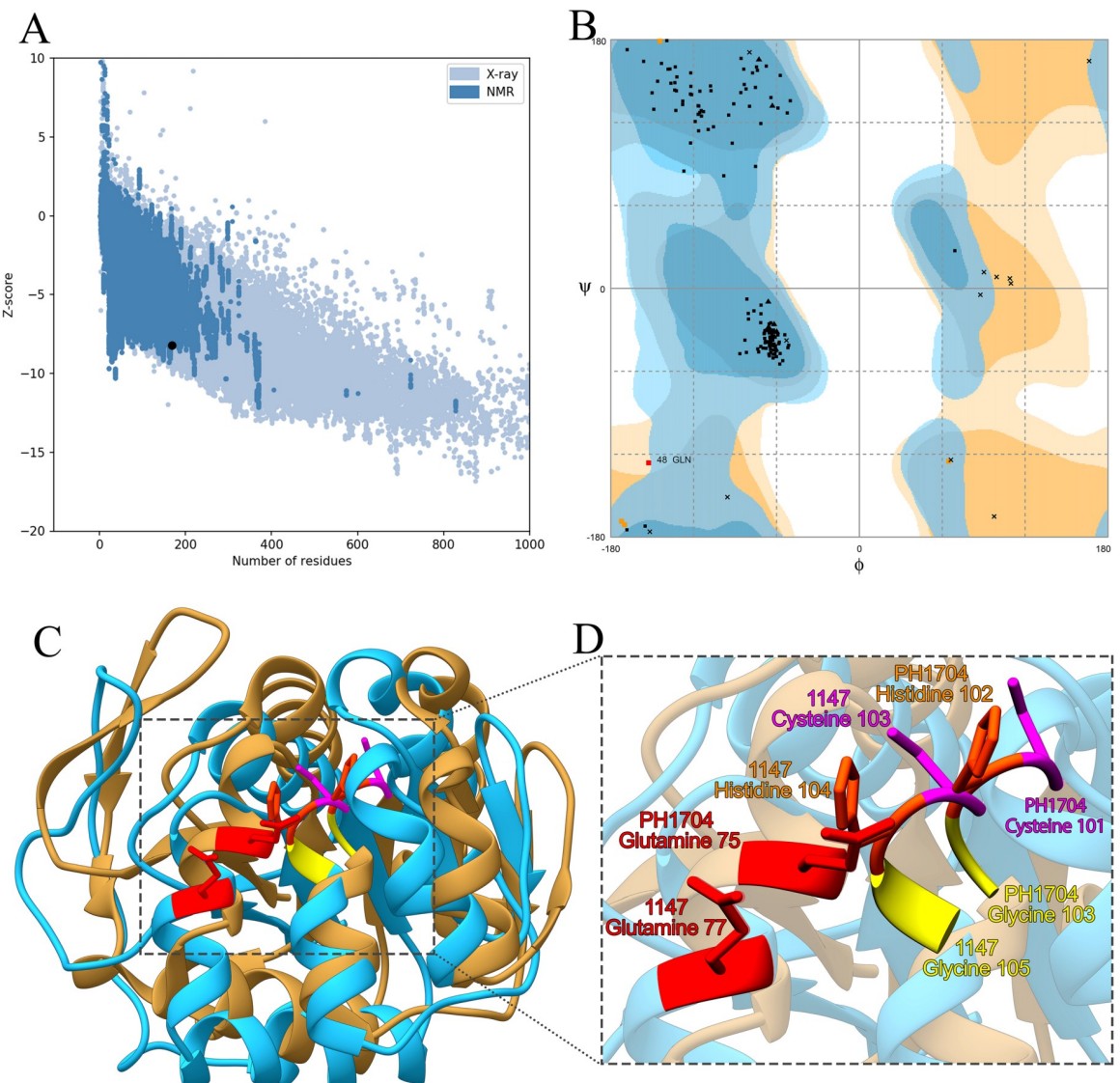

**Fig 3. The validation of homology modeling results.** (A) ProSA Z-score analysis for homology modeled protease 1147. The plot showed that the black spot (predicted model) with the Z-score value of -6.34 is within the range of native conformation of all proteins chains in PDB determined by X-ray crystallography or NMR spectroscopy concerning their length. (B) Ramachandran's analysis depicts that 98% of residues fall within the favoured, allowed, and acceptable regions, while the outlier region represents only 2%. (C) The 3D superimposition between the native structure of *Pyrococcus horikoshii* chain A (beige) and predicted protease 1147 structure (blue). The RMSD is 0.51 Å. (D) Catalytic tetrad residues are shown and labeled.

The predicted structure of 1147, together with the amino acids forming the active site of the enzyme (Glu[77]–Cys[103]–His[104]–Gly[105]), is shown in Fig 4.

Molecular docking between the active site of protease 1147 and Tween 20 ligand predicted a hydrogen bond interaction between the oxygen atom at position 6 of Tween 20 and oxygen atom at position 6 of Arg[162] with a distance of 3.10 Å. It was shown that after 40 ns of MD simulation, the predicted hydrogen bond was maintained (Fig 5A and 5B) and furthermore, a new hydrogen bond with a distance of 2.86 Å was formed between Tween 20 and Thr[164] in protease (Fig 5C and 5D). No hydrogen bond was identified between the SDS ligand and the active site of protease 1147. The only predicted hydrogen bond between SDS and the protease was

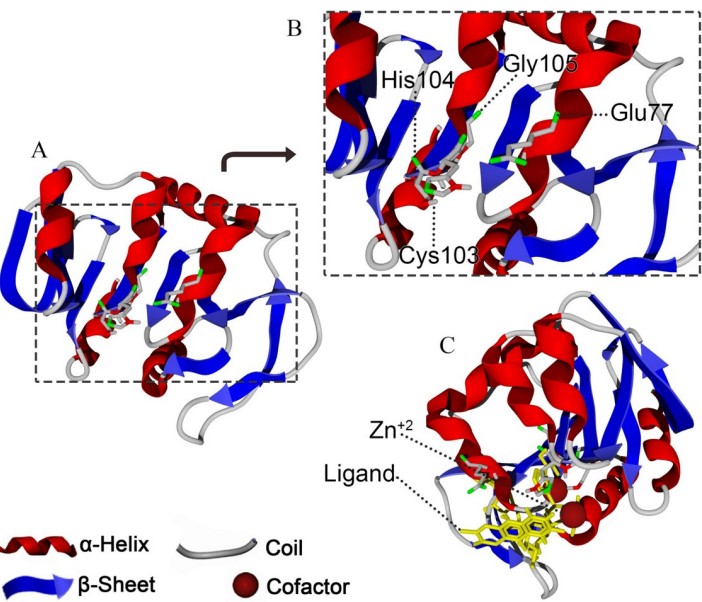

**Fig 4. The exhibition of protease 1147 binding site.** (A) Predicted 3D structure of protease 1147. (B) Amino acids belonging to the catalytic tetrad ($Glu^{77}$–$Cys^{103}$–$His^{104}$–$Gly^{105}$), and (C) nucleophilic elbow, enzyme cofactor, and active site flap in protease 1147 active site.

between oxygen atom at position 4 and $Lys^{19}$ in distance of 2.62 Å (Fig 6A and 6B). The predicted hydrogen bond was lost at the end of 40 ns MD simulation, and SDS was stripped off from the protein (Fig 6C and 6D).

The overall RMSD of the docked protease 1147-Tween 20 complex (0.23 nm) was in the lower value compared to the protease 1147 (0.39 nm) (Fig 7A). Root Mean Square Fluctuation (RMSF) plot indicated less structural fluctuation profile in the active site of protease 1147 in

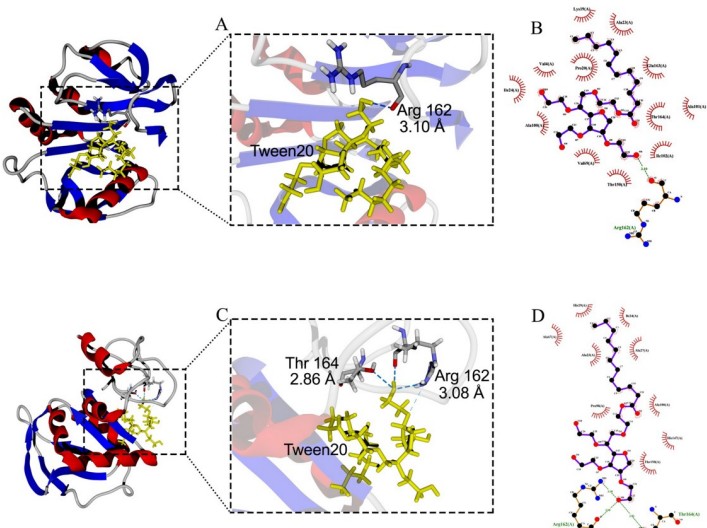

**Fig 5. Docking of protease 1147 and Tween ligand.** (A) Docked protease 1147-Tween 20 complex (before MD simulation), and (B) its 2-D display indicated an interaction between $Arg^{162}$ of protease 1147 with oxygen atom at position 6 of Tween 20. (C) Docked protease 1147-Tween 20 complex (after MD simulation), and (D) its 2-D display indicated two interactions between $Arg^{162}$ and $Thr^{164}$ of protease 1147 with Tween20.

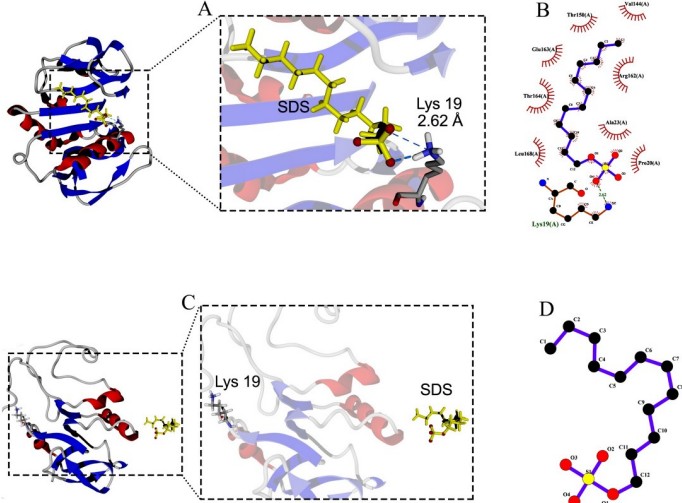

**Fig 6. Docking of protease 1147 and SDS ligand.** (A) Docked protease 1147-SDS complex indicated (before MD simulation), and (B) its 2-D display indicated an interaction between Lys[19] of protease 1147 with oxygen atom at position 4. (C) Docked protease 1147-SDS complex (after MD simulation), and (D) its 2-D display indicated loss of the interaction. The results of docking runs were completely identical.

complex with Tween 20 compared to the protease 1147 alone (Fig 7B). The radius of gyration (Rg) decreased in the protease 1147-Tween 20 complex, which indicates higher protease stability when it is complexed with Tween 20 (Fig 7C). The MD simulation of protease 1147 at 100˚C indicated protease instability after 20 ns and loss of structure after 40 ns.

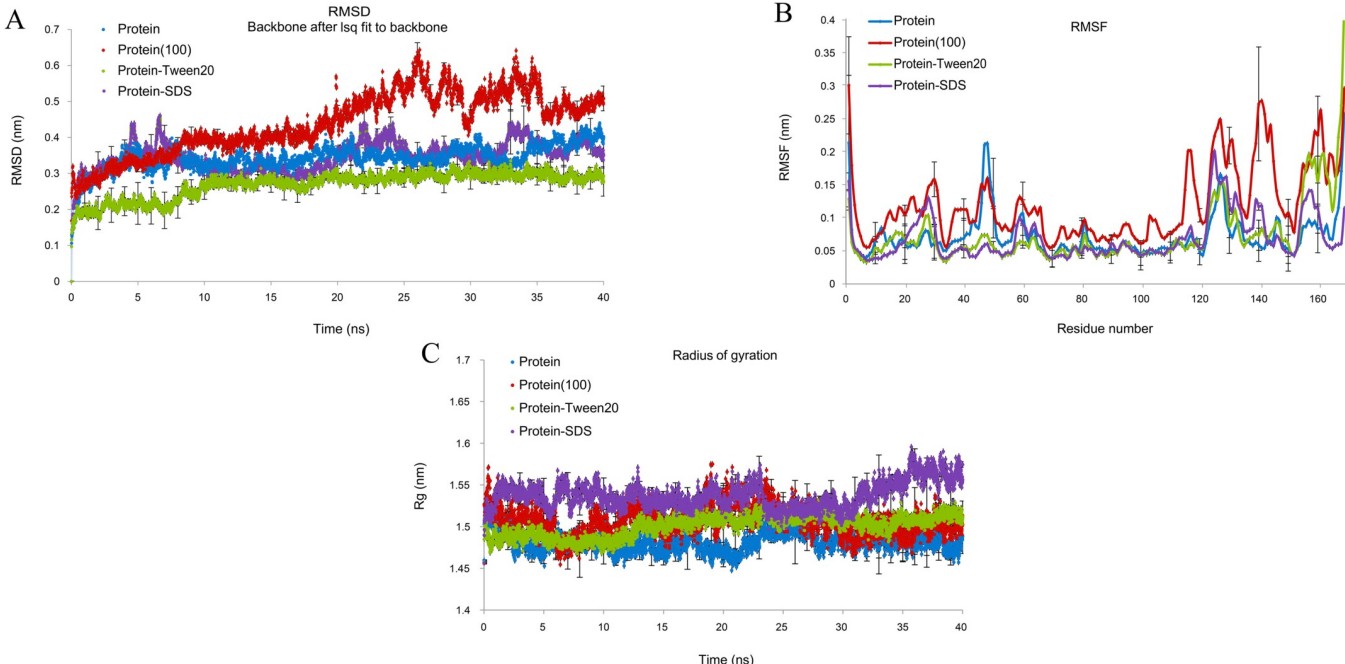

**Fig 7. MD simulation of protease 1147 alone and in complex with Tween 20 and SDS.** (A) RMSD graph, (B) RMSF graph, and (C) Rg Protease 1147 in the water environment (black), protease 1147-Tween20 complex (green), protease 1147-SDS complex (blue) and protease 1147 at 100˚C temperature. The approximate positions of active site amino acids are shown in the RMSF plot. The simulations were run in triplicate to obtain average values. The error bars represent standard deviations and are depicted for every 10 ps.

## Heterologous expression of protease 1147 in *E. coli*

The gene encoding protease 1147 from *Cohnella sp*. A01 genomic DNA was amplified using PCR and inserted into the expression plasmid pET26b(+) under the control of the T7 promoter. The accuracy of recombinant plasmid was authenticated using PCR with specific primers, double digestion, and sequencing (S3 Fig). This recombinant plasmid was used to transform *E.coli* BL21. The modified *E.coli* strain was cultivated in shake flasks. The SDS-PAGE analysis revealed intracellular accumulation of protein with a molecular weight of approximately 18 kDa and in a soluble form (Fig 8A).

## Purification, zymography and western blot analysis of soluble expressed protease 1147

The recombinant C-terminal His-tagged protease 1147 was purified effectively using Ni-NTA column and single-step thermal shock with overall yields of around 88% and 73%, respectively. The SDS-PAGE results of purification are shown in Fig 8A, and protein activity and efficiency of purification are summarized in Table 1. Casein substrate zymography showed the protease activity at an apparent molecular mass of 18 kDa (Fig 8B). Western blot analysis by anti-poly-Histidin peroxidase antibody revealed a similar mass for the recombinant protease 1147 (Fig 8C).

## Substrate specificity and kinetic measurement

Substrate specificity was inferred by comparing known protease substrates. The protease activity was higher when casein was used as substrate (100%), followed by gelatin (90%), azo-casein (25%), and BSA (10%). The kinetic parameters of protease 1147 were determined for the casein substrate by drawing the Michaelis-Menten curve (S4 Fig). The values of $K_m$, $k_{cat}$, and $k_{cat}/K_m$ was determined $13.72 \pm 0.6$ mM, $3.143 \times 10^{-3}$ (s$^{-1}$), and $0.381$ (M$^{-1}$ S$^{-1}$), respectively.

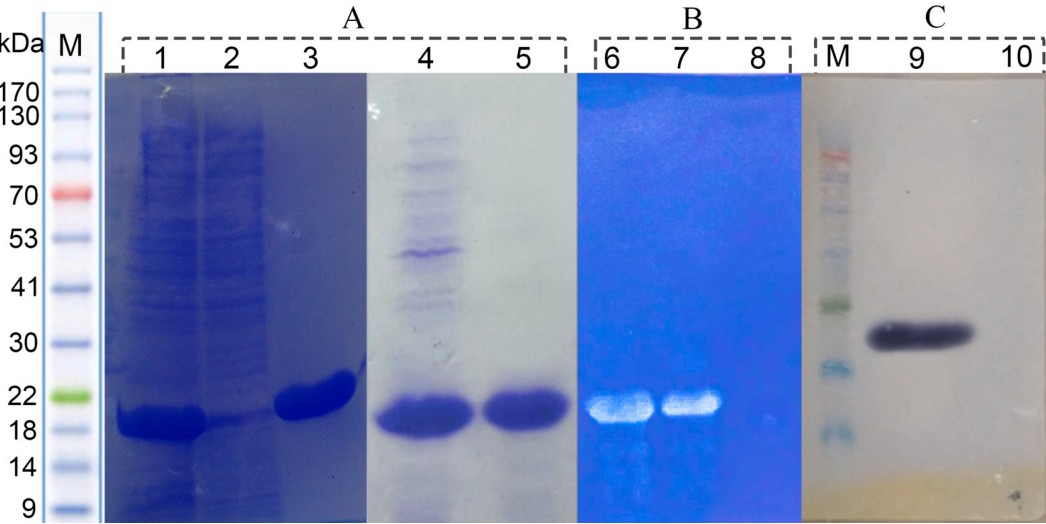

**Fig 8. SDS-PAGE, western blot and zymogram analysis of recombinant protease 1147.** (A) SDS-PAGE analysis of recombinant protease 1147. Lane 1 and 4) crude extract of recombinant BL21, lane2) crude extract of non-recombinant BL21 (negative control), lane 3) Ni-NTA purified, and lane 5) heat shock purified recombinant protease 1147. (B) Native-PAGE zymogram analysis of 6) Ni-NTA affinity column and, 7) heat shock single-step method purified protease 1147, 8) crude extract of non-recombinant BL21 (negative control). (C) Western blot analysis of (9) Ni-NTA affinity column purified protease 1147, and (10) negative control.

**Table 1. Protein activity and concentration, purified using Ni-NTA chromatography and single-step thermal shock.**

| Step | Volume (ml) | Total Activity (U) | Total Protein (mg) | Specific Activity (U/mg protein) | Yield (%) | Purification (fold) |
|------|-------------|-------------------|--------------------|----------------------------------|-----------|---------------------|
| Crude Extract | 50 | 14.1 | 49.2 | 0.28 | 100 | 1 |
| Ni-NTA | 4 | 12.4 | 0.7 | 17.7 | 87.9 | 62.3 |
| Thermal shock | 4 | 10.3 | 0.8 | 12.9 | 73.1 | 45.4 |

## High pH and thermal- stability of protease 1147

The maximum protease activity was obtained at 60°C (2.657 U/mg), and therefore, the optimal temperature was selected as 60°C (Fig 9A). The thermal stability of the recombinant enzyme is shown in Fig 9B. More than 80% of the protease activity remained after heat treatment at 70°C for 60 min. However, a sharp decrease was observed after that. The optimum pH was determined to be 7 for the protease 1147. The enzyme showed excellent pH stability in basic, neutral, and acidic pH (Fig 9C and 9D).

## Determination of protease 1147 activation energy and thermal inactivation parameters

The $E_a^{\ddagger}$ for protease 1147 was calculated 35.04 kJ/mol by Arrhenius equation (Fig 9E). $\Delta G^{\ddagger}$, $\Delta H^{\ddagger}$ and $\Delta S^{\ddagger}$ at optimum temperature were 74.36 and 32.68 kJ/mole, and 125 JmolK$^{-1}$,

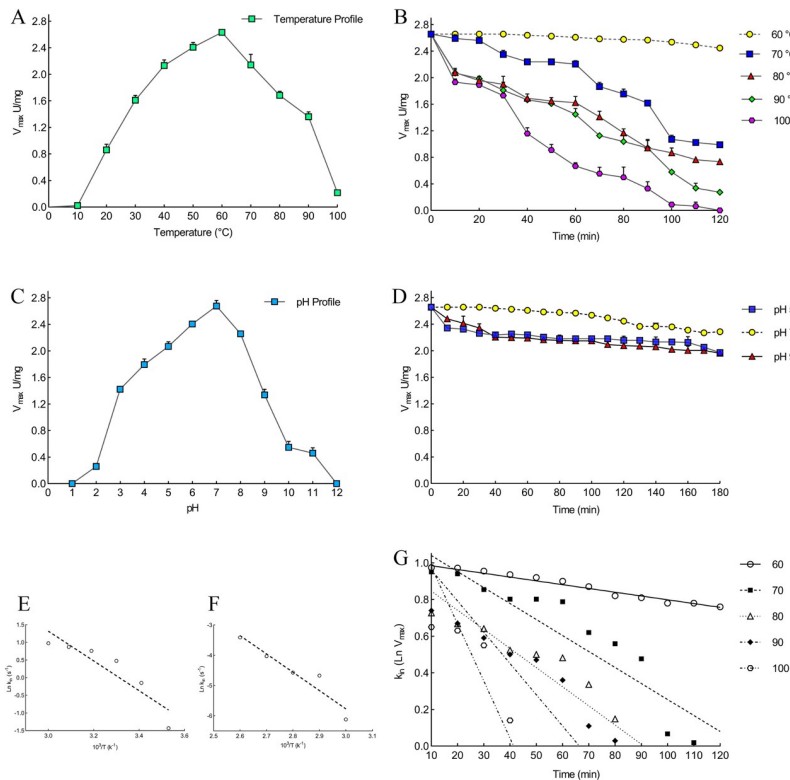

**Fig 9. Effect of temperature and pH on the recombinant protease 1147.** Effect of temperature on the recombinant protease 1147 (A) activity (B) stability. Effect of pH on the recombinant protease 1147 (C) activity (D) stability. The optimum temperature and pH were 60°C and 7, respectively. (E) Arrhenius plots for $E_a^{\ddagger}$. (F) Arrhenius plots for $E_a^{\#}$. (G) Thermal inactivation Ln at 60, 70, 80, 90, and 100°C.

respectively (Table 2). Lower values of $\Delta G^{\ddagger}$ and $E_a{}^{\ddagger}$ at optimum temperature revealed a higher stabilization and efficient transition state of the enzyme/substrate complex ($ES^{\ddagger}$) at 60˚C.

The obtained values for $\Delta G^{\ddagger}{}_{E-T}$ and $\Delta G^{\ddagger}{}_{E-S}$ (-2.66 and 0.19 kJ/mol, respectively) at 60˚C indicated that low amount of energy is required for protease 1147 to form the transition complex, and the negative $\Delta G^{\ddagger}{}_{E-T}$ indicates that the reaction occurs spontaneously.

The $k_{in}$ values obtained for protease 1147 indicated an increasing trend with increasing the temperature from 60 to 100˚C (Table 3). Using an irreversible thermal inactivation plot (Fig 9G), it is deduced that the protease 1147 has the highest $t_{1/2}$ at 60˚C, and increasing the temprature led to lower stability and $t_{1/2}$. Furthermore, the $k_{in,}$ and D value were increased and decreased with increasing the temperature, respectively (Table 3). According to the results, protease 1147 can maintain its thermal stability at high temperatures.

The $E_a{}^{\#}$, for protease 1147 inactivation was calculate 50.31 kJ/mol using Arrhenius Equation (Fig 9F). $\Delta H^{\#}$ and $\Delta S^{\#}$ for inactivation of protease 1147, at optimum temperature, were 47.54 kJ/mole$^{-1}$ and 154.70 JmolK$^{-1}$, respectively, and showed a decreasing trend with increasing the temperatures (Table 3). The data obtained for $\Delta S^{\#}$, $\Delta H^{\#,}$ and $\Delta G^{\#}$ corroborate the previous evidence showing the thermostable features of the enzyme. The calculated parameters of irreversible activation of protease 1147 are compared with those from other studies in Table 3.

## Effects of metal ions, inhibitors, detergents, organic solvent, and surfactants on proteolytic activity of protease 1147

None of the metal ions in the selected concentrations showed a significant effect on the measured protease activity (Table B in S1 File). The effects of other molecules, including inhibitors, surfactants, and organic solvents, are summarized in Table 4. In organic solvents, β-Mercaptoethanol in lower concentrations had the highest positive effect on the activity of protease 1147. Inhibitory effects of methanol, ethanol, and isopropanol on the protease activity were observed at 3% v/v, while lower concentrations of methanol showed significant incremental effects. Also, 1% and 2% v/v of glycerol and DMSO had substantial positive effects on protease activity, respectively.

## Monitoring temperature, pH, and surfactant-induced protease 1147 unfolding by fluorescence spectroscopy

The aromatic amino acids in the protein sequence are intrinsic fluorophores, which absorb light at approximately 280 nm and emit UV light. The results of fluorescence analysis in Fig

**Table 2. The thermodynamic parameters for protease 1147 reaction activation energy.**

| Parameters | Value |
|---|---|
| $V_{max}$ (U/mg) | 2.657 |
| $K_m$ (mM) | 13.72 |
| $k_{cat}$ ($10^{-3}$ s$^{-1}$) | 3.143 |
| $k_{cat}/K_m$ (M$^{-1}$ S$^{-1}$) | 0.381 |
| $E_a{}^{\ddagger}$ (kJ/mole) | 35.04 |
| $\Delta G^{\ddagger}$ (kJ/mole) | 74.36 |
| $\Delta H^{\ddagger}$ (kJ/mole) | 32.68 |
| $\Delta S^{\ddagger}$ (Jmolk$^{-1}$) | 125 |
| $\Delta G^{\ddagger}{}_{E-T}$ (kJ/mol) | -2.66 |
| $\Delta G^{\ddagger}{}_{E-S}$ (kJ/mol) | 0.19 |
| $K_a$ (1/$K_m$) | 0.07 |

**Table 3. Thermodynamic parameters for irreversible thermal inactivation of protease 1147.** The parameters were measured after 120 min incubation at various temperatures, and the values obtained at 60°C compared with other studied thermostable proteases.

| Parameters | $k_{in}$ (m$^{-1}$) | $t_{1/2}$ (min) | D value (min/m$^{-1}$) | $E_a^{\#}$ (kJ/mol) | $\Delta H^{\#}$ (kJ/mol) | $\Delta G^{\#}$ (kJ/mol) | $\Delta S^{\#}$ (JmolK$^{-1}$) | Ref. |
|---|---|---|---|---|---|---|---|---|
| Temperature | | | | | | | | This study |
| 60°C | $2.10 \times 10^{-3}$ | 330.07 | 1318.52 | 50.31 | 47.54 | 98.71 | 154.70 | |
| 70°C | $8.70 \times 10^{-3}$ | 79.67 | 327.82 | | 47.54 | 97.83 | 147.60 | |
| 80°C | $10.40 \times 10^{-3}$ | 66.65 | 282.23 | | 47.37 | 100.27 | 149.85 | |
| 90°C | $17.10 \times 10^{-3}$ | 40.53 | 176.51 | | 47.28 | 101.56 | 149.51 | |
| 100°C | $31.20 \times 10^{-3}$ | 22.27 | 99.40 | | 47.20 | 102.52 | 148.29 | |
| Other thermostable proteases | | | | | | | | |
| 60°C | $39 \times 10^{-3}$ | 7.95 (h) | 59.05 | 105.50 | 140.23 | 90.84 | 148.31 | [25] |
| 60°C | $12.40 \times 10^{-3}$ | 45 | - | 34.34 | 31.12 | 93.17 | -184.75 | [26] |
| 60°C | $4.41 \times 10^{-3}$ | 82.42 | 4.56 (h) | 20.96 | 6.36 | 92.23 | -0.26 | [27] |
| 60°C | $27.51 \times 10^{-3}$ | 25.19 | - | 91.80 | 111.26 | 113.92 | 57.03 | [28] |
| 60°C | $15.83 \times 10^{-3}$ | 19.62 | 155.6 | 187.4 | 184.63 | 68.15 | 349.79 | [29] |
| 60°C | $19.50 \times 10^{-3}$ | 35.54 | - | 93.81 | 91.14 | 92.76 | -8.39 | [30] |
| 60°C | $35 \times 10^{-3}$ | 19.80 | 65.80 | 37.19 | 34.42 | 102.53 | -204.43 | [31] |
| 60°C | $50 \times 10^{-3}$ | 34 | - | 69 | 66.98 | 90.2 | -69.5 | [32] |

**Table 4. Effects of various organic solvents, inhibitors, and surfactants on protease 1147 activity.** Each value represents the mean of three independent replicates with a standard deviation.

| Effector Molecule | Relative activity (%) | | |
|---|---|---|---|
| Organic solvent | | | |
| Control | 100 | | |
| Concentration (v/v) | **1%** | **2%** | **3%** |
| β-Mercapto | 150.6±2.3 | 143.3±2.1 | 126.9±15.4 |
| Methanol | 120.0±1.8 | 113.6±2.4 | 83.9±7.8 |
| Ethanol | 99.1±7.7 | 98.4±2.6 | 86.9±3.4 |
| Isopranol | 94.8±5.0 | 93.6±4.1 | 86.6±5.5 |
| Glyserol | 126.9±3.8 | 117.8±2.7 | 94.4±2.3 |
| Aceton | 92.1±5.2 | 93.9±3.8 | 90.6±1.4 |
| DMSO | 116.9±2.3 | 122.1±3.2 | 86.6±3.4 |
| Inhibitors | | | |
| Control | 100 | | |
| Concentration (v/v) | **1%** | | **2%** |
| GuHCl | 91.8±2.7 | | 82.1±1.3 |
| DTT | 95.7±3.4 | | 87.2±3.9 |
| PMSF | 79.3±3.4 | | 48.4±1.3 |
| IAA | 10.2±1.4 | | NA |
| E-64 | NA | | NA |
| Leupeptin | NA | | NA |
| EDTA | 97.8±2.2 | | 94.8±3.6 |
| Surfactants | | | |
| Control | 100 | | |
| Concentration (v/v) | **1%** | | **2%** |
| Tween 20 | 415.7±5.0 | | 350.3±5.0 |
| Tween 80 | 488.7±7.8 | | 460.0±19.5 |
| SDS | 100.0±1.7 | | 59.0±0.9 |

10A and 10B show that the maximum fluorescence emission was at 316 nm after excitation at 280 nm. The protease 1147 maintained its overall structure in temperatures lower than 70°C and pH of 4–8. While the addition of 1% SDS seems to cause a minor reduction in the fluorescence intensity, a substantial change was recorded after the addition of 1% Tween 20, which is due to the unfolding of protease 1147 tertiary structure (Fig 10C).

ANS fluorescence was used to analyse the exposure of hydrophobic patches in the protein. Fig 10D and 10E shows ANS spectra of protease 1147 in the various temperatures and pH. In temperatures lower than 80°C and pH between 4–12, the protease is well folded, and hydrophobic patches are not exposed. A large increase in the fluorescence strength was recorded for ANS outside of the temperature mentioned above and pH ranges, which is characteristic of protein unfolding. A striking increase in the fluorescence intensity was observed after the addition of 1% Tween 20, while 1% SDS caused slight changes in the fluorescence intensity (Fig 10F).

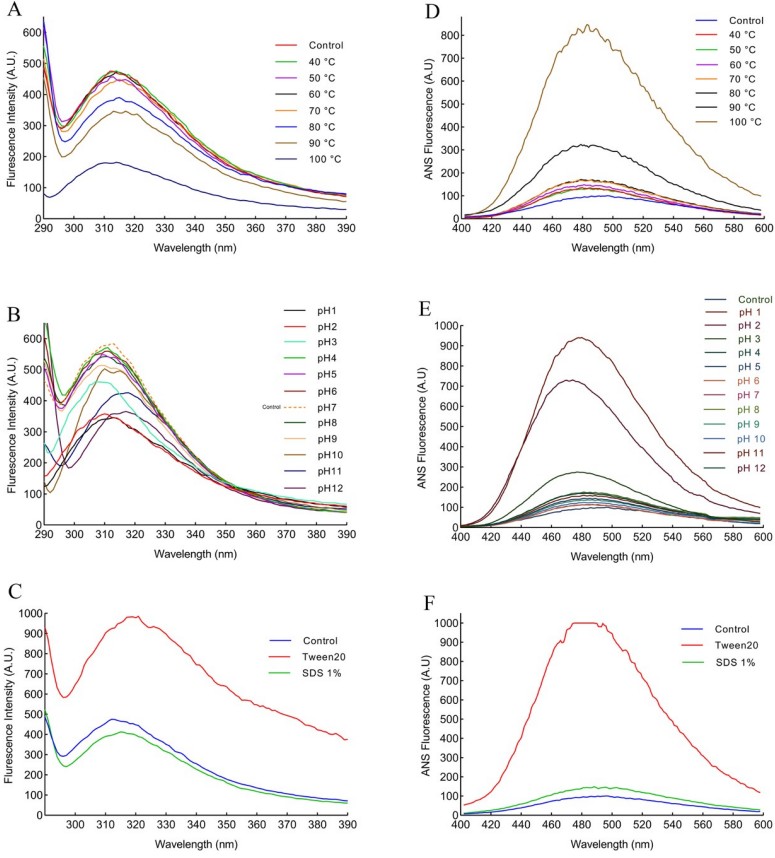

**Fig 10. Intrinsic and extrinsic fluorescence spectroscopy of the protease 1147.** (A) Intrinsic fluorescence spectroscopy of protease 1147 in the temperature range of 40–100°C exhibited a decrease in fluorescence intensity in temperature higher than 80°C and a blue shift in 100°C. (B) Intrinsic fluorescence spectroscopy of protease 1147 in the pH range of 1–12 showed fluorescence decline in pH higher than 9 and lower than 3. A blue and red shift was observed in acidic and basic pH, respectively. (C) Intrinsic fluorescence spectroscopy of protease 1147 in the presence of 1% Tween 20 and SDS, indicated a slight reduction of fluorescence intensity for SDS and a huge increase and red shift for Tween 20. (D) Extrinsic fluorescence spectroscopy of protease 1147 in the temperature range of 40–100°C exhibited a slight increase in fluorescence strength for temperature below 80°C and a significant increase for 90 and 100°C. (E) Extrinsic fluorescence spectroscopy of protease 1147 in the pH range of 1–12 showed huge fluctuation and blue shift of fluorescence strength in strongly acidic (pH 1–3) solution. (F) Extrinsic fluorescence spectroscopy of protease 1147 in the presence of 1% Tween 20 and SDS indicated a slight increase of fluorescence intensity for SDS and a huge increase for Tween 20.

## Determination of secondary structure changes by far-UV CD

The far-UV CD was performed to investigate protein folding and changes in the secondary structure of recombinant protease 1147 as a function of temperature, pH, and in the presence of Tween 20 and SDS. The secondary structure of purified protease 1147, at 25˚C and pH of 7, was considered as control, and the result showed the typical signature of folded proteins containing less α-helix and more β-sheet structures. The far-UV CD spectra were recorded at different temperatures, pH values, and in the presence of two chemical additives, i.e., Tween 20 and SDS (Fig 11). The secondary structures were calculated using software CDNN and listed in Table 5.

The highest degree of ellipticity change and disintegration for protease 1147 structure occurred at 100˚C and a wavelength of 220 nm. Furthermore, at this temperature, very intense disorganisation at 208 nm was observed compared to the control spectrum. As indicated in Table 5, the β-turn and α-helix levels increased by 0.2 nm and 0.7 nm at 100˚C, respectively, while the rate of β-sheet was dropped 0.9 nm. At very acidic and basic pH values, intense disorganisation occurred, and the protease 1147 began to lose its secondary structure. The secondary structure contents were not significantly altered in the presence of 1% SDS. While Tween 20 respectively caused an increase and a decrease in the α-helix and β-sheets values.

## Discussion

Because of significant industrial demands, there is a great interest in functional screening for novel proteases with improved characteristics. Due to several unique advantages such as large diversity, rapid growth, and small space required for cultivation, microbes (fungi and bacteria) are preferred sources for naturally-occurring proteases [33]. Compared to fungal proteases, bacterial proteases have higher reaction rates and better heat tolerance [34]. Several new, desired protease-encoding bacterial genes have been expressed in new hosts, with the aim of overproduction, characterization, and engineering [35, 36].

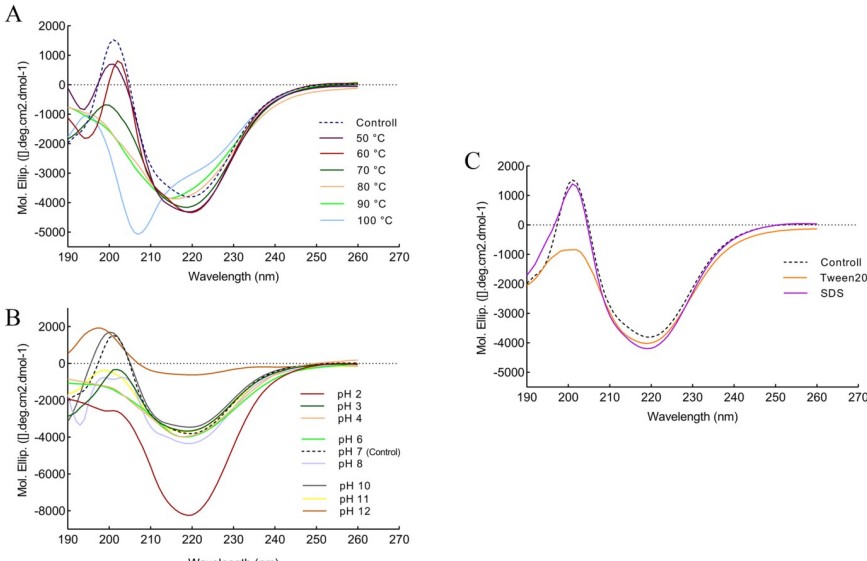

**Fig 11. Analaysis of secondary structure changes by far-UV CD.** Far-UV CD spectra of protease 1147, (A) at different temperatures, (B) at different pH values, and (C) in the presence of 1% Tween 20 and SDS. These spectra indicated the changes in the secondary structure content of protease 1147. The spectrum of purified protease 1147 at temperature 25˚C and pH of 7 was considered as control.

**Table 5. Effects of different temperatures, pH values, SDS, and Tween 20 on the protease 1147 secondary structure and protease activity.**

| Protease 1147 | α-helix (%) | β-sheets (%) | β-turn (%) | Random (%) | Activity (%) |
|---|---|---|---|---|---|
| 50˚C | 16.9 | 31.4 | 20.5 | 48.6 | 90 |
| 60˚C | 17.0 | 31.8 | 20.5 | 48.5 | 100 |
| 70˚C | 16.7 | 31.4 | 20.7 | 48.1 | 80 |
| 80˚C | 16.5 | 31.0 | 20.9 | 47.9 | 60 |
| 90˚C | 16.5 | 30.9 | 20.9 | 47.7 | 50 |
| 100˚C | 17.2 | 29.8 | 20.9 | 48.3 | 5 |
| pH 2 | 16.2 | 31.0 | 20.9 | 48.9 | NA |
| pH 3 | 16.0 | 31.4 | 20.4 | 50.2 | 50 |
| pH 4 | 16.4 | 31.6 | 20.8 | 48.3 | 70 |
| pH 6 | 16.3 | 32.3 | 20.9 | 48.5 | 90 |
| pH 7 | 16.5 | 32.5 | 20.7 | 49.3 | 100 |
| pH 8 | 16.7 | 32.1 | 20.7 | 48.3 | 80 |
| pH 10 | 16.2 | 31.8 | 20.8 | 49.7 | 20 |
| pH 11 | 16.5 | 31.2 | 20.7 | 49.3 | NA |
| pH 12 | 20.0 | 29.1 | 21.8 | 57.4 | NA |
| SDS | 16.4 | 32.5 | 20.8 | 48.6 | 100 |
| Tween 20 | 17.0 | 31.7 | 20.8 | 48.6 | 300 |

In this study, we recombinantly expressed, purified, and described the detailed structural and biochemical properties of a newly isolated protease 1147 from the thermophilic indigenous *Cohnella sp*. A0.1. In silico analysis of the translated sequence of protease 1147 demonstrated that it is produced intracellularly with a high aliphatic index, which is a characteristic of thermophilic bacterial proteins [37]. Similar to other related proteases that were listed in Table A in S1 File, the calculated negative GRAVY value indicated that this protein is non-polar [38]. Phylogenetic analysis reveals the high amino acid similarity between the protease 1147 and other thermostable intracellular cysteine proteases from DJ-1/ThiJ/PfpI superfamily. This superfamily has grown to a large family that has a representative in nearly all organisms. Despite closely-related 3D structures in their core fold, this superfamily has diverse functions including acting as chaperons, neuron protection, and glyoxalase III [39]. Although many members of this superfamily have been classified, few of them have been biochemically characterized [40].

The enzymatic activity of the protease 1147 was provided by the identified conserved catalytic tetrad consisting of Cys[103], His[104], Glu[105], and Gly[77], in which the thiol side chain of cysteine functions as a nucleophile during the initial stages of catalysis.

BLASTP analysis of the amino acid sequence indicated that protease 1147 exhibited the highest structural homology with the PH1704 protease from hyperthermophilic archaean *Pyrococcus horikoshii*, which is a member of DJ-1/ThiJ/PfpI and was identified as both aminopeptidase and endopeptidase [41]. The crystal structure of PH1704 at a 2.0-Å resolution was used as a template for the construction of the protease 1147 model. The homology model of the query was generated using MODELLER 9v7, and the quality of the modeled 3D structure was validated by the Ramachandran plot, with only 2% of residues classified as outliers. The structure was further verified using ProSA, and the calculated Z-score (-6.34) revealed a compatible value with the template and a negative balance for the potential energy of the entire structure. The RMSD value between protease 1147 and template was 0.51 Å, indicating a high structural similarity and good model structure.

The modeled building indicated that the protease 1147 has an overall α/β sandwich tertiary structure, which is present in the known structure of DJ-1/ThiJ/PfpI superfamily members

[39]. Furthermore, the motif of the nucleophilic elbow, containing the conserved Cyc[103] located at the very sharp turn between a β-strand and an α-helix, has been identified in the modeled structure. This motif is known to allow easy access to the cysteine by substrate [42]. The predicted 3D location and the topology of the active site in protease 1147 in the modeled structure has a strong match with known template PH1704.

Among different additives that were tested to evaluate the effect, it was shown that the protease activity was significantly enhanced in the presence of two non-ionic surfactants (Tween 20 and 80). Another ionic surfactant, SDS, at 1% concentration did not affect the enzyme activity. Interaction mechanisms of protease 1147 were investigated in the presence or absence of Tween 20 and SDS by molecular docking and MD. The results indicated that Tween 20 contacts the Arg[162] in the active site of protease 1147 by hydrogen bond in aqueous solutions. The interaction remained almost stable after 40 ns of simulation and strengthened with two new contacts between the NH1 of the same Arg[162] and Thr[164] with the Tween 20 molecule. The only hydrogen bond between the Lys[19] and SDS was diminished after the simulation time. This result explains why SDS did not affect the enzymatic activity of protease 1147.

The results of RMSD and RMSF and the reduced Rg emphasise that Tween 20 positively affects protein stability when complexed with the protease 1147. The plots of RMSD and RMSF were similar for protease 1147 alone and in complex with SDS. Tween 20 is a non-ionic surfactant commonly used in the formulation of biotherapeutic products to stabilise proteins and prevent protein aggregation and surface adsorption [43]. However, the mechanism of the stabilizing effect of Tween 20 and other non-ionic surfactants has not been fully explained. The dual nature of surfactants and their amphiphilic structure produce strong affinity for interphases, and this is the characteristic that is the basis of mechanisms by which surfactant affects the physicochemical properties of proteins [44, 45]. It is reported that, besides altering interactions of the protein with surfaces, Tween 20 acts as a chemical chaperone and aids in protein folding via direct hydrophobic interaction with proteins [46]. Other proteases with increased activity in the presence of Tween 20 are listed in Table C in S1 File.

Because protease 1147 has been obtained from thermophilic bacteria and maintained its activity up to 90˚C, the MD simulation was run at 100˚C to show the conformational changes. Significant differences were observed when comparing the RMSD and RMSF for the simulation at 100˚C. Greater RMSD and RMSF and increased Rg observed at 100˚C indicated that at this temperature, the protein started to denature, and because of a large amount of heat in the system, many bonds begin to break apart.

As shown in Fig 8, the recombinant protease 1147 was detected in the soluble form at approximately 18 kDa position and verified by zymography and Western blot analysis. The majority of bacterial proteases have a molecular weight ranging between 15 and 45 kDa [47]. The produced protein was purified using His-tag affinity column chromatography and a non-chromatographic purification method named single-step thermal shock. The latter method led to a lower purification fold and yield. However, this fast, low-cost, and convenient process makes it appropriate for use in laboratories and large-scale purification of protease 1147. The purified protease 1147, using both methods, was homogenous as it gave a single protein band on SDS-PAGE.

As we showed in this study, protease 1147 can tolerate a high range of temperatures. Thus, for the thermal shock method, we incubate the enzyme-containing supernatant at 90˚C for 10 min. This method has been used successfully for purification of thermostable proteins but in lower temperatures and for a longer time (30 min at 60–70˚C) [48, 49].

We subsequently characterized the purified recombinant protease 1147 biochemically, investigating the influence of pH, temperature, and different chemical additives on the enzyme activity. Casein was used as the substrate as protease 1147 displayed the highest specificity for

it. The enzyme displayed excellent tolerance to high temperatures and a broad range of pH, with the highest activity at 60°C and a pH of 7. At 90°C, the protease had almost 50% activity, but the enzyme activity significantly decreased at 100°C, likely due to the denaturation of the enzyme. Furthermore, MD simulations at 100°C confirmed this result. This high degree of thermal stability may be beneficial in many industrial processes, for example, to decrease the risk of contamination at high temperatures and the cost of external cooling [50]. Concerning the influence of pH, protease 1147 shows approximately 80% and 50% of relative activity in pH values of 5 and 9, respectively, with good stability after 3 h of incubation at 60°C. The activity and stability over such a broad pH range and temperatures make protease 1147 potentially useful for different industrial applications.

Studies on the thermodynamic parameters of the enzyme have provided a basic insights into the factors that determine the enzyme stability. As we have shown here, the protease 1147 was shown to has intrinsic structural stability and maintains its activity at high temperatures.

The activation energy ($E_a$) is the energy that must be provided for the reaction to occur. The reduction of $E_a$ barrier increases the fraction of reactant molecules that achieve sufficient energy to form a product. On the other hand, the maximum values of $\Delta H$ and $\Delta S$ suggested the effectiveness of the transitional state and negative $\Delta G$ measures the spontaneity of the catalytic reaction [26]. In this study, we performed a thermodynamic analysis to determine $E_a^{\ddagger}$, $\Delta G^{\ddagger}$, $\Delta H^{\ddagger}$, $\Delta S^{\ddagger}$, $\Delta G^{\ddagger}_{E-T}$, $\Delta G^{\ddagger}_{E-S}$, and $K_i$ for protease 1147. The low values of $E_a^{\ddagger}$, $\Delta G^{\ddagger}$, besides high values of $\Delta H^{\ddagger}$, $\Delta S^{\ddagger}$ at the optimum temperature indicated the efficient transition state of $ES^{\ddagger}$ at this temperature. These parameters were measured for thermostable proteases in other studies such as those obtained from *Bacillus stearothermophilus* [25], or *Aspergillus fumigatus* [32]. Compared to the mentioned studies, here, we found lower $\Delta G^{\ddagger}$, and $E_a^{\ddagger}$, and the lower amounts of $\Delta G^{\ddagger}_{E-T}$ compared to the $\Delta G^{\ddagger}_{E-S}$ in this study suggested that the reaction can occur spontaneously.

The parameters of irreversible inactivation are usually expressed in terms of $k_{in}$, $t_{1/2}$, and D value. Several studies have shown that $t_{1/2}$ and D value are high in heat-resistant enzymes [25, 26, 51]. Compared to other heat-resistant proteases, the acquired high $t_{1/2}$ and D value and notable low $k_{in}$ at 60°C confirmed the thermostability nature of the protease 1147. Moreover, the significant lower D value at the higher temperatures indicated its higher susceptibility to thermal inactivation in temperature above 60°C.

The activation energy for the thermal inactivation of an enzyme is a key factor in understanding its thermostability. Enzymes undergo a first-order inactivation reaction, which is responsible for irreversible denaturation. The high $E_a^{\#}$ for inactivation means more energy is needed for denaturation of the enzyme that indicated higher thermostability. The thermodynamic parameters of irreversible inactivation, including $E_a^{\#}$, $\Delta H^{\#}$, $\Delta G^{\#}$, and $\Delta S^{\#}$ were evaluated for protease 1147 to assess the $E_a^{\#}$. The high value of $\Delta S^{\#}$ reflects its conformational stability and resistance towards the denaturation process [52]. The values of $\Delta H^{\#}$ and $\Delta S^{\#}$ of the protease 1147 showed a decreasing trend in higher temperatures, indicating the changes in the conformation of this enzyme toward the partially unfolded transition state [25]. The unfolding of the enzyme structure is associated with an increase in disorder or entropy led to the lower $\Delta S^{\#}$. Furthermore, high measured amounts of $\Delta G^{\#}$ and $E_a^{\#}$, at the optimum temperature, revealed that protease 1147 resist against the unfolding of its transition state, and the enzyme required high inactivation energy to get denatured. However, increasing the temperature conversely affect the rigidity and thermostability of protease 1147. Overly, the findings indicate that the transition states of protease 1147 enzyme at optimum temperature was more ordered. These results, may provide a piece of evidence that the higher catalytic efficiency ($k_{cat}/K_m$) of the protease 1147 is due to the stability of the transition state [25].

Even though the bioinformatics analysis revealed that protease 1147 is most similar to the thermostable protease PH1704, it showed a different pattern of activity at the pH, and temperature ranges studied. It was shown that PH1704 has the highest activity at approximately 80˚C and a pH of 8.5 [41]. To compare its optimum pH, temperature, and other kinetic parameters with protease 1147, a summary of characterized bacterial proteases from different sources was provided in Table D in the S1 File.

Many studies have reported that metal ions impose a stabilizing effect on enzymes and are required to maintain the enzyme structure, enhance the enzyme activity, and protect it against thermal inactivation [53, 54]. Different metal ions have been reported to affect various proteases activity [54, 55]. However, a limitation in the use of cysteine proteases is that metal ions readily suppress their activity. Therefore, these proteases require mild reductants and chelating agents and thus are not economical. As such, serine proteases that do not have this limitation may be preferred for industrial applications [56]. The effect of nine kinds of metal ions was determined at optimum pH and temperature, and the result showed that unlike many other cysteine proteases, protease 1147 is almost stable in different concentrations of metal ions with just a slight decrease in the proteolytic activity. According to the predicted structure, $Zn^{2+}$ is needed as a co-factor to incorporate in the ligand-binding site of protease 1147. However, 1, 2, and 5 mM concentrations of $Zn^{2+}$ did not affect the protease activity.

The protease 1147 was found to be fairly stable in the presence of studied organic solvents, and its activity enhanced by β-mercaptoethanol and lower concentrations of methanol, glycerol, and DMSO. This result is quite remarkable, considering that the technological utility of an enzyme can be greatly expanded if an enzyme can perform its activity in the presence of organic solvents rather than just an entirely aqueous reaction media [57, 58].

The protease 1147 enzyme activity was completely inhibited by IAM, E-64, and leupeptin, which are the typical cysteine protease inhibitors. However, metalloprotease inhibitor EDTA and denaturant agent GuHCl used conventionally in protease inhibitor cocktails did not have a significant inhibitory effect against protease 1147. PMSF, the serine/cysteine protease inhibitor used in a 2% concentrated solution, inhibits the protease activity up to approximately 50%. These results validated that this protease belongs to the cysteine protease family.

Fluorescence spectroscopy and CD analysis of proteins give a complete picture of the overall structure and are generally used to monitor the conformational changes of proteins with changes in the solvent composition or the environment [59]. Intrinsic fluorescence emission due to the tyrosine and tryptophan residue provides a sensitive reporter to characterize the protein. These residues are very sensitive to the polarity of the environment [60]. The intrinsic fluorescence emission of protease 1147 at a pH of 7 and 60˚C displayed an emission maximum at 316 nm, suggesting that at least some of 5 tyrosine residues in the protease 1147 were exposed to the solvent. Increasing the temperature up to 80˚C with pH in the range of 3 to 9 caused a small decrease in the fluorescence intensity, whereas at 100˚C and in higher and lower pH values a sharp decrease in the fluorescence occurred.

The observed decrease in emission wavelength indicated that these environmental changes cause tyrosine residues to become more buried in the hydrophobic core of the enzyme, leading to the distinct structural changes. Upon the addition of 1 mM Tween 20, the emission spectra showed a 2-fold increase in intensity. This increase is likely due to the exposure of tyrosine residues. The tyrosine residues are exposed to the solvent in the unfolded state of proteins [61]. A surfactant such as Tween 20 has an anti-aggregation effect and increases the free energy of unfolding, exposing the protein side chains to the surrounding [61]. The wavelength bands of protease 1147 in the presence of 1% SDS showed an approximately 10% decrease in intensity. It was reported that a low concentration of SDS can promote protein aggregation [62], which can cause tryptophan residues to remain

inaccessible. However, our results suggested negligible perturbation of protease 1147 structure in the presence of SDS 1%. Conformational changes in protease 1147 have also been studied by binding of fluorescence probe ANS. The results closely resemble those of the intrinsic fluoresce. A steep increase in fluorescence intensity was observed at 100˚C, very basic and acidic pH, and in the presence of Tween 20. These results strongly suggest that protease 1147 under these conditions attains a conformation where hydrophobic patches are exposed on the surface. The protein at pH 7 is in its native compact form and thus the hydrophobic regions are out of reach. At pH 1 and 2 the highest fluorescence emission was detected, and a slight blue shift (4 nm) at pH 3 may be due to the aggregation of protein in acidic phase. The decrease in fluorescence intensity at pH 3 may be due to the presence of the molten globule (MG) intermediate state in the protease 1147. Similarly, lipase-3646 [24] and cysteine protease ZCPG [63] were shown that exhibit the behavior of the MG state at pH 3 and pH 2, respectively. The overall results were in agreement with those obtained for temperature and pH stability and the MD simulation in the presence of selected surfactants.

The far-UV CD was used to generalize the secondary structure of protease 1147 [64]. The CD spectrum shows a positive CD band near 200 nm and a negative one at 220 nm, which was consistent with the reported CD spectrum of Der p1 cysteine protease from douse dust, and papain from the latex of Carica papaya [65, 66]. The percentage of secondary structure was calculated as 17% α-helix, 31.8% β-sheet, and 20.5% β-turn. As the temperature increased, the ellipticity value considerably reduced, suggesting the loss of regular secondary structures. When the temperature was further raised to more than 80˚C, the positive peak at 200 nm almost vanished. Disruption of structures in very basic and acidic pH values was also detected. At a pH of 2, a marked increase in the negative band was observed. Tween 20 causes a considerable decrease in the negative band at 200 nm, and the use of SDS shows no sign of change in the secondary structure of protease 1147. Overall, the high structural stability of protease 1147 is likely due to its high content of β-sheet, making it less susceptible to unfolding [67].

## Supporting information

**S1 Fig. SignalP data.** SignalP software suggested no signal peptide for protease 1147.
(PPTX)

**S2 Fig. Residues energy data.** ProSA local model quality/residue-wise energy plot shows most of the residues have negative energy.
(PPTX)

**S3 Fig. Cloning analyses.** M, DNA marker; 1, empty pET26b(+) vector; 2, recombinant pET26b(+) containing protease 1147 sequence; 3, double digestion of recombinant pET26b(+) by XhoI and NdeI restriction enzyme; 4 negative control PCR; 5, PCR from recombinant pET26b(+) vector with specific primers for protease 1147 sequence.
(PPTX)

**S4 Fig. Michaelis-Menten data.** Michaelis-Menten plot of protease 1147 activity as a function of casein concentrations.
(PPTX)

**S1 File.** Table A) Biochemical properties of protease 1147 and closely related proteases. NA, number of amino acids; MW, molecular weight; pI, theoretical isoelectric point; GRAVY, grand average of hydropathy; AI, aliphatic index; II, instability index. Table B) Effects of various metal ions on protease 1147 activity. Table C) The effect of metal ions, chemical compounds, inhibitors and surfactants on the activity of some other proteases. Table D) Type,

Temperature, pH, substrate specificity and kinetic characterization of some other proteases. (DOCX)

**S1 Raw images.**
(PDF)

## Acknowledgments

The authors would like to thank the National Institute of Genetic Engineering and Biotechnology (NIGEB) for providing the research facility by grant number 971215-I-712.

## Author Contributions

**Conceptualization:** Saeed Aminzadeh.

**Data curation:** Hossein Tarrahimofrad, Ehsan Jahangirian, Somayyeh Rahimnahal.

**Formal analysis:** Hossein Tarrahimofrad, Amir Meimandipour, Sareh Arjmand, Mohammadtaghi Beigi Nassiri.

**Investigation:** Hossein Tarrahimofrad, Ehsan Jahangirian, Somayyeh Rahimnahal.

**Methodology:** Hossein Tarrahimofrad.

**Project administration:** Saeed Aminzadeh.

**Software:** Hossein Tarrahimofrad, Javad Zamani.

**Supervision:** Mohammadtaghi Beigi Nassiri, Saeed Aminzadeh.

**Validation:** Hossein Tarrahimofrad, Amir Meimandipour, Sareh Arjmand, Mohammadtaghi Beigi Nassiri.

**Visualization:** Hossein Tavana, Javad Zamani.

**Writing – original draft:** Hossein Tarrahimofrad, Amir Meimandipour.

**Writing – review & editing:** Sareh Arjmand, Hossein Tavana.

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
