## [Decision Letter · Decision Letter 0]

26 Nov 2019

PONE-D-19-30576

Structural and Biochemical Characterization of a Novel Thermophilic Coh01147 Protease

PLOS ONE

Dear Dr. Aminzadeh,

Thank you for submitting your manuscript to PLOS ONE. After careful consideration, we feel that it has merit but does not fully meet PLOS ONE’s publication criteria as it currently stands. Therefore, we invite you to submit a revised version of the manuscript that addresses the points raised during the review process.

Dear Dr Saeed Aminzadeh

The manuscript has been reviewd by three independed experts in the filed and based on thier jugdment the paper need intense improvment. The major issues that reviewer rised are:

1. Paper need improvment in purification and charatceristaion section.

2. The whole paper needs significant re-writing to improve clarity of the data 

3. Molecular dynamics section requires additional experiments or comments from your side.

We would appreciate receiving your revised manuscript by Jan 10 2020 11:59PM. To enhance the reproducibility of your results, we recommend that if applicable you deposit your laboratory protocols in protocols.io, where a protocol can be assigned its own identifier (DOI) such that it can be cited independently in the future. For instructions see: http://journals.plos.org/plosone/s/submission-guidelines#loc-laboratory-protocols

We look forward to receiving your revised manuscript.

Kind regards,

Adam Lesner

Academic Editor

PLOS ONE

Journal Requirements:

1. Please include captions for your Supporting Information files at the end of your manuscript, and update any in-text citations to match accordingly. Please see our Supporting Information guidelines for more information: http://journals.plos.org/plosone/s/supporting-information.

Reviewers' comments:

Reviewer's Responses to Questions

**Comments to the Author**

1. Is the manuscript technically sound, and do the data support the conclusions?

Reviewer #1: Yes

Reviewer #2: Partly

Reviewer #3: Yes

2. Has the statistical analysis been performed appropriately and rigorously? 

Reviewer #1: Yes

Reviewer #2: No

Reviewer #3: Yes

3. Have the authors made all data underlying the findings in their manuscript fully available?

Reviewer #1: Yes

Reviewer #2: No

Reviewer #3: Yes

4. Is the manuscript presented in an intelligible fashion and written in standard English?

Reviewer #1: Yes

Reviewer #2: Yes

Reviewer #3: Yes

5. Review Comments to the Author

Reviewer #1: Major:

1. May I suggest to modify all the subtitles in Result section to be more “result – orientated”. In other words, not on what the experiments have been done, but what the results obtained.

2. Fig 5, For the MD simulation and docking results, can authors comment on how repeatable / robust they are? In other words, will docking / MD generate other simulations if run multiple times?

Minors:

3. Abstract: Kcat to kcat

4. Page 12, line 7, what is the “TCA solution”?

5. Table 1, isoelectric point is pI not Pi.

6. Page 19, line 7, please spell out RMSD.

7. Fig 7b, y-axis label is missing

8. Fig 6, May consider moving it to SI since less significant

9. Tables 2 and 3, the significant digits – two digits should be enough. And no way to achieve 4 digits there.

Reviewer #2: Aminzadeh and coworkers describe a structural and biochemical characterization of a novel protease from thermophilic Coh01147 bacteria (Cohnella sp.). The structural analysis involves sequence homology analysis to identify a related known structure, and then homology modeling to produce a model of the enzyme. Molecular dynamics simulations are used to suggest a specific complex formation with surfactant Tween 20. A web-based server is used to predict a number of biophysical parameters for the protein. The protease is cloned and expressed and purified using a single-step His-tag strategy. Enzyme activity is characterized using zymography, temperature-dependent activity measurements, and a Michaelis-Menten analysis using casein. Other fluorescence spectroscopy studies were reported, including ANS binding; temperature and pH effects upon fluorescence and CD were also reported.

There are a number of criticisms of the report (in no particular order):

1. The authors report kcat as "Kcat", suggesting it is an equilibrium constant instead of the correct rate constant nomenclature

2. In several instances the authors report 4-5 digits of precision for values. This level of precision would seem unsupported

3. The interaction with Tween20 appears somewhat inconsistent. The authors report molecular dynamics studies show specific Tween 20 binding, but can only provide a single H-bond for such interaction. They also report Tween20 increases enzyme activity, but then also report that unfolding is induced by addition of Tween20.

4. Enzyme assays are potentially problematic for proteases that might self-digest (thereby reducing enzyme concentration over time). The authors do not address this possibility.

5. The Michaelis-Menten study utilizes casein as a substrate. There is likely more than one site of cleavage with such a large and complex polypeptide, but the MM treatment assumes a single substrate (i.e. site of cleavage). The authors may be fitting experimental data to an inappropriate model

6. In the purification the authors report a mass of imidazole in the elution buffer, but no concentration. It is therefore impossible to repeat the study with the data provided

7. The authors perform a NiNTA purification step, and then follow it with a heat incubation step. However, in their purification table the specific activity decreases significantly after the heat incubation step. Normally the highest specific activity material is utilized for biochemical study. Why did the authors settle upon a lower specific activity material?

8. The authors quote the ProtParam web server reported instability index as "confirming" the protease structure is stable. Such analyses confirm nothing, they only suggest

9. The authors state that ANS binding is related to exposure of hydrophobic patches. More generally, ANS binding is considered to indicate molten globule structure

10. The authors report spectroscopic data as a function of temperature and pH. In principle, such data can be deconvoluted (e.g. with two-state, reversible denaturation models) to fit thermodynamic parameters of stability (i.e. DeltaG of unfolding, and so on). The authors however do not carry their analysis through to this point. In viewing the data presented, it is possible that unfolding is not two-state or cooperative (but this is speculation in the absence of any such analysis).

11. In general, a number of figures or tables are not so significant and could be put into supplemental data. In other words, the authors fill the report with figures and tables of questionable significance.

Reviewer #3: The manuscript by Tarrahimofrad et al. describes the production and characterization of a thermostable protease. While the topic of the manuscript is interesting, and the enzyme described may find some commercial application, there are multiple issues that need to be addressed/clarified.

1. Page 5, lines 21-22

This sentence is confusing, as it suggests that the gene is thermostable and not the protein.

2. Page 7, line 16

Please add information on sequence identity and similarity between P. horikoshii protease and the protease 1147.

3. Page 8, line 7

The title of the paragraph should be rather “Protein-ligand biding – docking studies”.

4. Page 10, line 4

There is “bacterial colon” and it should read “bacterial colony”.

5. Paragraph on recombinant protein purification

In buffers’ formulations the imidazole concentration should be reported instead of milligrams quantities of the compound.

6. To reference various manuscripts the authors use “et al” and it should read “et al.”.

7. Table 1 can be moved to the Supplementary Materials

Please add the unit for MW.

There is “Pi” and should be “pI”.

8. Page 19, line 7

Unit for the reported RMSD value is missing.

9. Generally all tables are poorly looking and some work has to be done to make them more esthetically pleasing.

10. Please comment on the possible influence of the polyhistidine tag on the enzymatic activity of the protease.

11. Please provide information on the quaternary form of the protein. In the case of the protease from P. horikoshii only the oligomeric form of the enzyme was active.

12. Please use “kcat” instead of “Kcat”.

13. Table 3

Please correct the way how the errors are reported. For example, there is “91.81±2.7” and it should be “91.8±2.7”. Please check the error reported for SDS, as currently it is zero.

14. Table 4

Please report the percentage values of of the secondary structure without decimals.

How do these values compare with the percentage of the secondary structure calculated based on the protein model?

15. Please report KM and Vmax values together with their experimental errors.

6. PLOS authors have the option to publish the peer review history of their article (what does this mean?). If published, this will include your full peer review and any attached files.

Reviewer #1: No

Reviewer #2: No

Reviewer #3: No

---

## [Author Response · Author response to Decision Letter 0]

12 Feb 2020

Journal: PLOS ONE 

Manuscript #: PONE-D-19-30576

Title of Paper: Structural and Biochemical Characterization of a Novel Thermophilic 

 Coh01147 Protease 

Authors: Hossein Tarrahimofrad, Amir Meimandipour, Sareh Arjmand, Mohammadtaghi Beigi Nassiri, Ehsan Jahangirian, Hossein Tavana, Javad Zamani, Somayyeh Rahimnahal, Saeed Aminzadeh

Dear Dr. Adam Lesner

We appreciate the time and efforts by the editor and referees in reviewing this manuscript. We have addressed all issues indicated in the review report carefully and believed that the revised version can meet the journal publication requirements. 

Yours sincerely 

Saeed Aminzadeh 

Reviewer #1:

 Major:

1. May I suggest to modify all the subtitles in Result section to be more “result – orientated”. In other words, not on what the experiments have been done, but what the results obtained.

We appreciate the Reviewer’s careful reading of the manuscript. The subtitles in the Results section were modified to be more “result-oriented”. The modified subtitles were written in blue font. 

2. Fig 5, For the MD simulation and docking results, can authors comment on how repeatable/robust they are? In other words, will docking/MD generate other simulations if run multiple times?

To ensure that the results are repeatable, Docking, and MD simulation were repeated three times. For this purpose, the ligands and the modeled protease 1147 were prepared. Using Molegro Virtual Docker (MVD) software, protein-ligand dockings were performed between protease 1147 and SDS as well as protease 1147 and Tween20. The molecular dynamics simulation were repeated as well. All dynamic molecular factors such as temperature and pressure adjustment as well as sample topology files were performed according to the initial run. Repeating the experiments indicated that the results are reproducible. 

Performing the experiments in triplicate was noted in the materials and methods, and footnote of figures 6 and 7 (in red font), and figure 7 was updated. 

Minors:

3. Abstract: Kcat to kcat

The word Kcat was replaced with kcat.

4. Page 12, line 7, what is the “TCA solution”?

TCA stands for Trichloroacetic acid. The abbreviation was introduced in the manuscript. 

5. Table 1, isoelectric point is pI not Pi.

The typo was corrected, and table 1 moved to supplementary file. 

6. Page 19, line 7, please spell out RMSD.

It was spelled out in the materials and methods, section; homology modeling and validation.

7. Fig 7b, y-axis label is missing

The y-axis label was added in the RMSD graph.

8. Fig 6, May consider moving it to SI since less significant

We feel this image can help in the proper retrieval of data because it shows the loss of connectivity between the 1147-SDS complex at the end of the MD. Because, as explained in the manuscript, at the beginning of molecular dynamics simulation, SDS establishes a hydrogen bond with the protein. This hydrogen bond was established with the amino acid Lysine 19 of the protein shown in Figure 6 (a and b). However, at the end of the molecular dynamics simulation, the hydrogen bond between SDS and the amino acid lysine 19 is lost, as shown in part c and d of Figure 6. In Overall, Figure 6 shows the loss of the hydrogen bond between SDS and protease 1147. In Compared with Figure 5, which shows the maintenance and enhancement of hydrogen bonding, Fig 6 can show the difference in the interaction at the beginning and end of the molecular dynamics simulation between the protein and the ligand.

9. Tables 2 and 3, the significant digits – two digits should be enough. And no way to achieve 4 digits there.

Thanks for the comment. The digits were corrected. 

Reviewer #2: 

Aminzadeh and coworkers describe a structural and biochemical characterization of a novel protease from thermophilic Coh01147 bacteria (Cohnella sp.). The structural analysis involves sequence homology analysis to identify a related known structure, and then homology modeling to produce a model of the enzyme. Molecular dynamics simulations are used to suggest a specific complex formation with surfactant Tween 20. A web-based server is used to predict a number of biophysical parameters for the protein. The protease is cloned and expressed and purified using a single-step His-tag strategy. Enzyme activity is characterized using zymography, temperature-dependent activity measurements, and a Michaelis-Menten analysis using casein. Other fluorescence spectroscopy studies were reported, including ANS binding; temperature and pH effects upon fluorescence and CD were also reported. There are a number of criticisms of the report (in no particular order):

1. The authors report kcat as "Kcat", suggesting it is an equilibrium constant instead of the correct rate constant nomenclature

Thanks for the comment. The word Kcat was replaced with kcat in the text. 

2. In several instances the authors report 4-5 digits of precision for values. This level of precision would seem unsupported

Thanks for the comment. The digints were corrected.

3. The interaction with Tween20 appears somewhat inconsistent. The authors report molecular dynamics studies show specific Tween 20 binding, but can only provide a single H-bond for such interaction. They also report Tween20 increases enzyme activity, but then also report that unfolding is induced by addition of Tween20.

Thanks for the comment. This is a very important question. MD results showed that the values of RMSD, RMSF, and Rg of the tween 20-1147 complex were much lower than that of protease 1147. Even the number of hydrogen bonds increased at the end of the molecular dynamics. These results confirm that the binding of tween 20 enhances the stability of the protein 1147 structure.

Several articles reported that heat-resistant proteins generally have a compact structure. This compact structure allows the protein to withstand temperature and pH changes and maintain its structure. This property is also transmitted to the active site of the enzyme, and the amino acids in the active site are located in a tight space in the protein structure. Tween 20 is a surfactant and can act as an oil and increase the flexibility of the protein structure. Tween 20 affects the active site of the enzyme, facilitating the opening of the active site and entry of the substrate. In fact, Tween 20 converts the compact structure of the protein into a flexible structure.

So far, there have been no reports of the mechanism of action of Tween 20 on protein, but in this study we concluded that Tween 20 can reduce protein structure compactness and makes the active site’s amino acids more readily linked to the substrate and increases the activity of the enzyme. This increase in activity is due to the fact that tween 20 changes the protein structure. This structural changes caused variations in the wavelength of the spectrophotometric studies. 

Furthuremore, to ensure that the results are repeatable, Docking, and MD simulation were repeated three times and figure 7 was updated. 

4. Enzyme assays are potentially problematic for proteases that might self-digest (thereby reducing enzyme concentration over time). The authors do not address this possibility.

The protease used in this study came from a thermophilic source. In general, enzymes from thermophilic sources have a compact structure. Because hydrophobic amino acids make up the bulk of the protein. This helps to increase the protein's ability to withstand high temperatures and maintain structure against pH changes. This compact structure allows protease 1147 to endure against autolysis conditions. To ensure about the lack of self-digestion, we subjected the recombinant protease 1147 in three different times (one day, one week and two weeks after expression ) on the SDS-PAGE. The result indicated in figure 1R. According to this qualificative result, the autolysis can not be problametic in our experiments. We should add that we used the recombinant protease 1147 one day after expression and exactly after purification. Furthermore, all steps of purification were performed at at 4 °C or using a cold chain (ice).

Figure R1. SDS-PAGE of recombinant protease 1147 1: One day after expression, 2: One week after expression, 3: Two week after expression. M:: Protein marker.

5. The Michaelis-Menten study utilizes casein as a substrate. There is likely more than one site of cleavage with such a large and complex polypeptide, but the MM treatment assumes a single substrate (i.e. site of cleavage). The authors may be fitting experimental data to an inappropriate model

We understand the term MM treatment in your question, as Molecular modeling (MM). in this case molecular modeling (MM) treatment and in vitro studies have shown that enzyme 1147 acts as an active domain of protease function. Protein-protein docking was not performed. We did not do molecular modeling research for the casein as substrate. We just performed Tween20 and SDS as ligand for molecular modeling and did not performed substrate (Casein).

6. In the purification the authors report a mass of imidazole in the elution buffer, but no concentration. It is therefore impossible to repeat the study with the data provided

Page 10, Lines of 13, 19, 20 and 21: The concentration of imidazole was written in mM of lysis buffer.

7. The authors perform a NiNTA purification step, and then follow it with a heat incubation step. However, in their purification table the specific activity decreases significantly after the heat incubation step. Normally the highest specific activity material is utilized for biochemical study. Why did the authors settle upon a lower specific activity material?

In this study, we used a purification step using a Ni Sepharose column (NiNTA) and purified the protein 1147 by the presence of the histidine tag. We also used a heat shock step due to the temperature stability of the protein 1147, instead of Ni Sepharose (NiNTA) column. Our rationale for using this heat shock method as well as reporting and comparing it with Ni Sepharose (NiNTA) method was that in general, purification processes are time-consuming and costly, especially if protein production is required in the large scale (such as pilot and even industrial).

If we can reduce the time to purification and the number of steps, then we can increase the cost-efficiency. If this protein needs to be purified on an industrial scale, the use of a heat shock purification step can greatly save cost and time.

The use of heat shock reduced the enzyme activity by 10% in this study, but if the enzyme needs to be produced and purified on a larger scale, the net efficiency, cost reduction, time and number of purification steps can be justified.

8. The authors quote the ProtParam web server reported instability index as "confirming" the protease structure is stable. Such analyses confirm nothing, they only suggest

Given that protein 1147 is inherently a thermophilic bacterium, we used data from the online software Protparam to show that factors related to the thermal properties of a protein were also found in protease 1147. However, we decided to move Table 1 to the supplementary file.

9. The authors state that ANS binding is related to exposure of hydrophobic patches. More generally, ANS binding is considered to indicate molten globule structure

 Thank you for the comment. We agree with the refree. A MG phase is observed. We provide further explanations in the following paragraph. We also mentioned this in the manuscript in green font (page 31 in discussion session).

Using ANS binding to protein investigated at 40 to 100 °C showed that fluorescence emission occurred at temperatures at which the hydrophobic regions of the protein were exposed to this solvent. Therefore, it was concluded that the compact structure of the enzyme 1147 is formed at 60 °C because due to inaccessibility of ANS to the internal domain and lack of binding. The maximum binding of ANS to the enzyme was at 100 °C, which could be due to the loss of protein folding and hydrophobic groups exposed to the solvent at this temperature. The maximum fluorescence emission from 60 °C to 100 °C was a blue-shift from 480 nm to 474 nm. The decrease of fluorescence that occurs as a result of the inability of the ANS binding to the hydrophobic region indicates that the enzyme maintains its structure from 40 to 90 °C.

The enzyme at pH 1and 2exhibited the highest fluorescence emission. When the ANS cannot bind to the hydrophobic region of the enzyme, fluorescence emission does not occur. The closer to the optimum pH of the enzyme, the lower the binding rate of ANS to the enzyme. Compared to the natural protein structure (pH 7), when the pH was reduced to 3, a slight blue shift (4 nm) was observed in maximum fluorescence emission, which may indicate the presence of intermediate state (MG). The intensity of the ANS fluorescence spectrum is consistent with the activity of protease 1147 in the presence of casein substrates. 

10. The authors report spectroscopic data as a function of temperature and pH. In principle, such data can be deconvoluted (e.g. with two-state, reversible denaturation models) to fit thermodynamic parameters of stability (i.e. DeltaG of unfolding, and so on). The authors however do not carry their analysis through to this point. In viewing the data presented, it is possible that unfolding is not two-state or cooperative (but this is speculation in the absence of any such analysis).

It is assumed that irreversible denaturation of proteins is a two-step reaction:

N ↔ U→ I

N is the native protein form, U is the partially unfolded protein state and I is the irreversible inactivated protein form.

https://doi.org/10.1038/s41598-019-55587-9

11. In general, a number of figures or tables are not so significant and could be put into supplemental data. In other words, the authors fill the report with figures and tables of questionable significance.

We improved some of the Figures(Fig.s 1, 2, 3, 7), table 1 was moved to supplementary and the tables 1, 2, and 3 were updated. 

Reviewer #3:

The manuscript by Tarrahimofrad et al. describes the production and characterization of a thermostable protease. While the topic of the manuscript is interesting, and the enzyme described may find some commercial application, there are multiple issues that need to be addressed/clarified.

1. Page 5, lines 21-22

This sentence is confusing, as it suggests that the gene is thermostable and not the protein.

 The sentence is changed to “The goal of the present study was recombinant expression of a previously isolated thermostable protease gene (Coh01144)”.

2. Page 7, line 16 

Please add information on sequence identity and similarity between P. horikoshii protease and the protease 1147.

This comparison is shown in Figure 2, where the amino acid sequence of Protease 1147 and P. horikoshii has been compared and the conserved sequences between these two proteins and several other similar proteins to Protease 1147 have been identified. (In Figure 2, part A; the first and the last sequence belong to protease 1147 and P. horikoshii, respectively). 

The 3D superimposition between the native structure of Pyrococcus horikoshii chain A and predicted protease 1147 structure was shown in fig 3C.

3. Page 8, line 7

The title of the paragraph should be rather “Protein-ligand biding – docking studies”.

The title was changed to “Protein-ligand docking studies” and specified in blue font.

4. Page 10, line 4

There is “bacterial colon” and it should read “bacterial colony”.

Thanks for the comment. The typo was corrected.

5. Paragraph on recombinant protein purification

In buffers’ formulations the imidazole concentration should be reported instead of milligrams quantities of the compound.

Page 9, Lines of 16 snd 23; Page 10, Lines of 1 and 2: The concentration of imidazole was written in mM

6. To reference various manuscripts the authors use “et al” and it should read “et al.”.

Page 11, Line 6 and 20: Both cases were corrected in the manuscript.

7. Table 1 can be moved to the Supplementary Materials

Please add the unit for MW.

There is “Pi” and should be “pI”.

The MW unit was added to the manuscript and Pi was also corrected and pI replaced. The table be moved to Supplementary. 

8. Page 19, line 7

Unit for the reported RMSD value is missing.

Unit for RMSD value (nm) was added to the graph and text.

9. Generally all tables are poorly looking and some work has to be done to make them more esthetically pleasing.

The updated tables were uploaded. 

10. Please comment on the possible influence of the polyhistidine tag on the enzymatic activity of the protease.

According to the bioinformatics analysis the amino acids of the active site of the enzyme 1147 are located in the middle of the protein sequence (Glu77, Cys103, His104, and Gly105). Since the histidin tag was incorporated at the C-terminal position of the enzyme, it has a logical distance that one can expects no interference. Furthermore the result of docking with Tween-20 and SDS did not show the establish of hydrogen bonding with the C-terminal his-tag 

To be more assure about the possible influence of his-tag, we modeled the structure with and without his-tag and superimpose them. As it is shown in the figure R3, no structural changes are observable. 

The long distance between the amino acids of active site and his-tag is shown in figure R3. 

Figure R2. The superimposition of 1147 with polyhistidine tag (Beige color) and 1147 without polyhistidine tag (cyan color). The polyhistidine tag residues showed in red color.

Figure R3. The apparent distance between the amino acids of the protease active site 1147 and the histidine tag. For clarity, the amino acids of the histidine tag are named.

11. Please provide information on the quaternary form of the protein. In the case of the protease from P. horikoshii only the oligomeric form of the enzyme was active.

Protein 1147 after Blast-PDB showed the most similarity to P. horikoshii protein A chain. Therefore, modeling homology was performed based on the PDB structure of P. horikoshii protein. Although P. horikoshii has been reported to be active in its oligomeric form, Protease 1147 was able to function in its monomeric form. Identification and determination of monomer-to-decamer protein forms require the use of advanced crystallographic techniques such as X-Ray and NMR. These methods are accurate and efficient. However, it takes a lot of time and cost. At this point, we used the integration of software and a server to predict the quaternary form of Protease 1147. Initially using MODELLER software, modeling of protease 1147 on horikoshii, in addition to chain A, protease 1147 structure on chain B and chain C of protease horikoshii were also performed. This structure was represented by Chimera. 

In addition to this step, we also predicted the fourth structure of protease 1147 based on the PDB structure of P. horikoshii protein in two forms of Asymmetric Unit and Biological Assembly 1 by the ZDOCK server. In this server, by selecting and introducing the binding site in all three chains, the protein structure was obtained in the Biological Assembly form and the protease 1147 quaternary form. The results related to the prediction of forth structure of protease 1147 are shown in figures R4-10.

Figure R4. The Cohnella sp. A01 protease active domain in Asymmetric Unit form.

Figure R5. The crystal structure of intracellular protease from Pyrococcus Horikoshii PH1704 in Asymmetric Unit form.

Figure 6. The superimposition of Cohnella sp. A01 protease 1147 and Pyrococcus Horikoshii PH1704 in Asymmetric Unit form.

Figure 7. The Cohnella sp. A01 protease 1147 quaternary chematic.

Figure 8. The superimposition of Cohnella sp. A01 protease 1147 and Pyrococcus Horikoshii PH1704 quaternary chematic.

Figure 9. The crystal structure of intracellular protease from Pyrococcus Horikoshii PH1704 in Biological Assembly form.

Figure 10. The superimposition of Cohnella sp. A01 protease 1147 and Pyrococcus Horikoshii PH1704 in Biological Assembly form.

12. Please use “kcat” instead of “Kcat”.

Thanks for the comment. The word Kcat was replaced with kcat in the text. 

13. Table 3

Please correct the way how the errors are reported. For example, there is “91.81±2.7” and it should be “91.8±2.7”. Please check the error reported for SDS, as currently it is zero.

Thanks for the comment. The data were corrected. 

14. Table 4

Please report the percentage values of the secondary structure without decimals.

How do these values compare with the percentage of the secondary structure calculated based on the protein model?

We reported the percentage values of the secondary structure with one decimals. We used bioinformatics softwares to predict the secondary structure. However, in this study, we calculated, in real and in vitro terms, the extent of the protease1147 structure using Circular Dichroism. We have the secondary structure information after 40 ns, whereas for spectroscopic studies, the enzymatic assay conditions were maintained for 15 minutes, after which the secondary structures were read. Therefore, secondary structure information is not comparable.

15. Please report KM and Vmax values together with their experimental errors.

Thanks for the comment. We reported Km and Vmax values with their experimental errors.

---

## [Decision Letter · Decision Letter 1]

31 Mar 2020

PONE-D-19-30576R1

Structural and biochemical characterization of a novel thermophilic Coh01147 protease

PLOS ONE

Dear Dr. Aminzadeh,

Thank you for submitting your manuscript to PLOS ONE. After careful consideration, we feel that it has merit but does not fully meet PLOS ONE’s publication criteria as it currently stands. Therefore, we invite you to submit a revised version of the manuscript that addresses the points raised during the review process.

We would appreciate receiving your revised manuscript by May 15 2020 11:59PM. To enhance the reproducibility of your results, we recommend that if applicable you deposit your laboratory protocols in protocols.io, where a protocol can be assigned its own identifier (DOI) such that it can be cited independently in the future. For instructions see: http://journals.plos.org/plosone/s/submission-guidelines#loc-laboratory-protocols

We look forward to receiving your revised manuscript.

Kind regards,

Paulo Lee Ho, Ph.D.

Academic Editor

PLOS ONE

Reviewers' comments:

Reviewer's Responses to Questions

**Comments to the Author**

1. If the authors have adequately addressed your comments raised in a previous round of review and you feel that this manuscript is now acceptable for publication, you may indicate that here to bypass the “Comments to the Author” section, enter your conflict of interest statement in the “Confidential to Editor” section, and submit your "Accept" recommendation.

Reviewer #1: All comments have been addressed

Reviewer #3: All comments have been addressed

Reviewer #4: (No Response)

2. Is the manuscript technically sound, and do the data support the conclusions?

Reviewer #1: Yes

Reviewer #3: Yes

Reviewer #4: Yes

3. Has the statistical analysis been performed appropriately and rigorously? 

Reviewer #1: Yes

Reviewer #3: Yes

Reviewer #4: Yes

4. Have the authors made all data underlying the findings in their manuscript fully available?

Reviewer #1: Yes

Reviewer #3: Yes

Reviewer #4: Yes

5. Is the manuscript presented in an intelligible fashion and written in standard English?

Reviewer #1: Yes

Reviewer #3: Yes

Reviewer #4: Yes

6. Review Comments to the Author

Reviewer #1: (No Response)

Reviewer #3: Please check the unit for KM. It reads "mg/ml-1" and it should be "mg/ml". All previously listed issues were addressed.

Reviewer #4: The revised version of the manuscript entitled “Structural and biochemical characterization of a novel thermophilic Coh01147 protease” (numbered: PONE-D-19-30576R1) has been well improved versus its original form, and it can be published in the journal PLOS ONE.

The above decision is based on that, generally, authors have responded carefully in most of the queries of authors, and were conformed to their constructive comments. On the other hand, and in my opinion, some responses of the authors to the comments of the reviewers are not satisfactory. Additionally, authors should take seriously into account more issues (see below) in their future publications.

In more details:

(a) Reviewer #2, comment #5: I think that the core of this question is related to the fact that authors did not used, as substrate, some relatively low molecular weight synthetic peptide specific for cysteine proteases (there are a plethora of them on the market), instead of casein (a protein). Authors’ responses “…enzyme 1147 acts as an active domain of protease function…”, along with “…We just performed Tween20 and SDS as ligand for molecular modeling and did not performed substrate….”, are confusing. Personally, I would strongly suggest to authors the use, in their future works, low molecular weight synthetic peptides, as proteases’ substrates; these latter offer to the experimenter much more information, especially in kinetic measurements (e.g. in the estimation of kcat, Km, kcat/Km, in pH- and temperature- profiles, etc).

(b) Reviewer #2, comment #10: It seems that more likely authors misunderstood the reviewer’s query and comments; in the cited address “https://doi.org/10.1038/s41598-019-55587-9” is referred a similar speculation. In my opinion these comments of Reviewer #2 are correct.

(c) Additional: Authors based some of their results on false and/or old-fashion, and statistically erroneous, treatments of their experimental kinetic data (original and revised versions of the manuscript).

(I) Authors estimated the M.M. parameters (kcat or Vmax, and Km) using the Lineweaver-Burk plot, i.e. the statistically most erroneous method; it has been rejected since 1961 (G.N. Wilkinson, The Biochemical journal, 80, 1961, 324–332). Nowadays, are easily available numerous statistically robust non-linear computer programs/algorithms, based also on non-parametric statistics, which are the specific ones for data from enzyme kinetic experiments. It is too pity those authors, who handle more complicated computer subroutines, to ignore and/or to underestimate important issues; this sounds unpleasant for the reputation of the authors!

(II) Figure 9: as ordinate in all four diagrams is the “Relative activity %”, which is meaningless and worthless according to the Current IUBMB recommendations on enzyme nomenclature and kinetics (A. Cornish-Bowden, Perspectives in Science 1, 2014, 74-87). The accepted meaningful entities, which should be used as ordinates in similar diagrams, are the M.M parameters kcat or Vmax, Km, kcat/Km or Vmax/Km. Furthermore, the pH-profiles of the aforementioned entities (Fig 9C) should be fitted using proper equations (e.g. ref. E.M. Papamichael et al, “Enzyme Kinetics and Modeling of Enzymatic Systems”, in ADVANCES IN ENZYME TECHNOLOGY, 1st Edition, p. 83/Eqs 3.11-3.14, and ref. A. Foukis et al, Bioresource Technology 123, 2012, 214–220/Eqs 1 and 1a), and estimate all important pKa-values. Likewise, the temperature profiles (Fig 9A) in all cases should be depicted by using the absolute scale as abscissa. Subsequently, the experimental data should be successively fitted by variants of the Arrhenius’ and Eyring equations (e.g. ref. A. Foukis et al, Bioresource Technology 123, 2012, 214–220/Eqs 2 and 3). In the case of the absolute temperature profiles, the activation thermodynamic parameters �G‡, ��H‡ and �S‡ could be estimated easily and more precisely (e.g. ref. E.M. Papamichael et al, “Enzyme Kinetics and Modeling of Enzymatic Systems”, in ADVANCES IN ENZYME TECHNOLOGY, 1st Edition, p. 83/Eqs 3.17-3.18/3.15-3.16).

7. PLOS authors have the option to publish the peer review history of their article (what does this mean?). If published, this will include your full peer review and any attached files.

Reviewer #1: No

Reviewer #3: No

Reviewer #4: Yes: Emmanuel M. Papamichael

---

## [Author Response · Author response to Decision Letter 1]

15 May 2020

Journal: PLOS ONE 

Manuscript #: PONE-D-19-30576R1

Title of Paper: Structural and Biochemical Characterization of a Novel Thermophilic 

 Coh01147 Protease 

Authors: Hossein Tarrahimofrad, Amir Meimandipour, Sareh Arjmand, Mohammadtaghi Beigi Nassiri, Ehsan Jahangirian, Hossein Tavana, Javad Zamani, Somayyeh Rahimnahal, Saeed Aminzadeh

Dear Dr. Paulo Lee Ho

We appreciate the time and efforts by the editor and referees in reviewing this manuscript. We have addressed all issues indicated in the review report carefully and believed that the revised.

On your recommendation, we have recorded a number of our own materials and methods in https://www.protocols.io/ and put the DOI of each in the manuscript.

We while revisied our submission, uploaded our all figure files to the Preflight Analysis and Conversion Engine (PACE) digital diagnostic tool, https://pacev2.apexcovantage.com/

Yours sincerely 

Saeed Aminzadeh 

We appreciate the reviewers who took the time to read the manuscript and helped us improve the manuscript with their comments. We have carefully applied the valuable comments of the esteemed reviewers in the manuscript.

Reviewer #3: Please check the unit for KM. It reads "mg/ml-1" and it should be "mg/ml". All previously listed issues were addressed.

The Km unit was corrected and expressed as “mM”. This unit was obtained using Prism GraphPad V.8 software and addressed in all previously listed issues:

Page 2, line 14.

Page 23, line 4 and 5.

Page 24, table 2.

Supplementary, S1 file, page 7, table D.

Reviewer #4: The revised version of the manuscript entitled “Structural and biochemical characterization of a novel thermophilic Coh01147 protease” (numbered: PONE-D-19-30576R1) has been well improved versus its original form, and it can be published in the journal PLOS ONE.

The above decision is based on that, generally, authors have responded carefully in most of the queries of authors, and were conformed to their constructive comments. On the other hand, and in my opinion, some responses of the authors to the comments of the reviewers are not satisfactory. Additionally, authors should take seriously into account more issues (see below) in their future publications.

In more details:

(a) Reviewer #2, comment #5: I think that the core of this question is related to the fact that authors did not used, as substrate, some relatively low molecular weight synthetic peptide specific for cysteine proteases (there are a plethora of them on the market), instead of casein (a protein). Authors’ responses “…enzyme 1147 acts as an active domain of protease function…”, along with “…We just performed Tween20 and SDS as ligand for molecular modeling and did not performed substrate….”, are confusing. Personally, I would strongly suggest to authors the use, in their future works, low molecular weight synthetic peptides, as proteases’ substrates; these latter offer to the experimenter much more information, especially in kinetic measurements (e.g. in the estimation of kcat, Km, kcat/Km, in pH- and temperature- profiles, etc).

We thank the referee for the explanation. You are absolutely right. As you said, unfortunately, we misunderstood the referee's intent and answered it based on a misunderstanding of the question.

We also thank you for your advice. Understanding your advice, we decided to use small synthetic substrates instead of casein in our future studies.

(b) Reviewer #2, comment #10: It seems that more likely authors misunderstood the reviewer’s query and comments; in the cited address “https://doi.org/10.1038/s41598-019-55587-9” is referred a similar speculation. In my opinion these comments of Reviewer #2 are correct.

Proteins in monomer mode are considered as two states. It is assumed that irreversible denaturation of proteins is a two-step reaction:

N ↔ U→ I

N is the protein native state, U is the reversibly and partially unfolded enzyme form and I is the enzyme irreversible inactivated state. The transition state (Tn*) which is formed between N and U is determining the irreversible thermodynamic parameters [1, 2].

Considering that we have considered the enzyme form monomer in all stages of bioinformatics and laboratory experiments by default. Therefore, when we have the monomer form of the protein, the protein can be considered as two states.

In addition, in the laboratory, we used PAGE with a non-regenerative form when examining the activity of the enzyme with a zymogram test. If the enzyme is a monomer, the protein will be in its true size, but if the protein is in the form of a dimer or multimeter, the protein will be in the area above its molecular weight. The 1147 protease was found in the non-regenerative PAGE of the zyogram, in front of its molecular weight about of 18.5 kDa (Fig 8B, in manuscript). 

The above allows us to consider the enzyme protease 1147 as a monomer protein.

ΔG# is the value usually shows the protein denaturation reaction. ΔH# and ΔS# are determining respectively the heat and entropy alteration in conversion reaction of native to the transition state. To verify this statement, we calculated the thermodynamic parameter of the irreversible deactivation energy and considered it as a function of the thermodynamic performance (ΔG# of unfolding) of the protease 1147 instade of deconvolve CD data with two-state, irreversible denaturation models. 

(c) Additional: Authors based some of their results on false and/or old-fashion, and statistically erroneous, treatments of their experimental kinetic data (original and revised versions of the manuscript).

We recalculated the kinetic data using the software and adjusted the other results based on the new calculations.

(I) Authors estimated the M.M. parameters (kcat or Vmax, and Km) using the Lineweaver-Burk plot, i.e. the statistically most erroneous method; it has been rejected since 1961 (G.N. Wilkinson, The Biochemical journal, 80, 1961, 324–332). Nowadays, are easily available numerous statistically robust non-linear computer programs/algorithms, based also on non-parametric statistics, which are the specific ones for data from enzyme kinetic experiments. It is too pity those authors, who handle more complicated computer subroutines, to ignore and/or to underestimate important issues; this sounds unpleasant for the reputation of the authors!

We read the reference introduced by you and we follow your advice. We recalculated all the enzyme kinetic arguments based on statistical nonlinear computer programs and algorithms using prism software, and we put them in the manuscript and corrected all the results based on the new M.M parameters.

Page 11, line 16 and 17.

Page 23, line 4 and 5.

Supplementary, S4 Fig.

(II) Figure 9: as ordinate in all four diagrams is the “Relative activity %”, which is meaningless and worthless according to the Current IUBMB recommendations on enzyme nomenclature and kinetics (A. Cornish-Bowden, Perspectives in Science 1, 2014, 74-87). The accepted meaningful entities, which should be used as ordinates in similar diagrams, are the M.M parameters kcat or Vmax, Km, kcat/Km or Vmax/Km.

All diagrams related activity were corrected and reported based on the Vmax parameter:

Figure 9 A, B, C & D.

Furthermore, the pH-profiles of the aforementioned entities (Fig 9C) should be fitted using proper equations (e.g. ref. E.M. Papamichael et al, “Enzyme Kinetics and Modeling of Enzymatic Systems”, in ADVANCES IN ENZYME TECHNOLOGY, 1st Edition, p. 83/Eqs 3.11-3.14, and ref. A. Foukis et al, Bioresource Technology 123, 2012, 214–220/Eqs 1 and 1a), and estimate all-important pKa-values. Likewise, the temperature profiles (Fig 9A) in all cases should be depicted by using the absolute scale as abscissa.Subsequently, the experimental data should be successively fitted by variants of the Arrhenius’ and Eyring equations (e.g. ref. A. Foukis et al, Bioresource Technology 123, 2012, 214–220/Eqs 2 and 3).

Thank you very much for your guidance. We've read the top-level references you've introduced, and we've come up with some very important new ones.

We are pleased to announce that we are currently studying the 1147 Protease Crystallography, and we plan to use the tips and equations you have introduced in our next article, along with information on crystallography and low molecular weight synthetic substrate.

Also, as the amount of content presented in our current study increases, we plan to use and report your proposed equations in our next study. 

In the case of the absolute temperature profiles, the activation thermodynamic parameters �G‡, ��H‡ and �S‡ could be estimated easily and more precisely (e.g. ref. E.M. Papamichael et al, “Enzyme Kinetics and Modeling of Enzymatic Systems”, in ADVANCES IN ENZYME TECHNOLOGY, 1st Edition, p. 83/Eqs 3.17-3.18/3.15-3.16).

Thank you very much. We used the appropriate equation (Eyring) introduced in your book and calculated the data for estimating the energy activation of the enzyme and other thermodynamic parameters related to the conversion of the enzyme to the product. Based on this, we brought the relevant data in the materials and methods section, the results, as well as the discussion. In this way, this data was updated in two tables and one section added in Photo 9. After completing and replying to your comment, we have identified the all items required in the manuscript:

Material and methods: Page 13, line 17 to page 16, line 12.

Resultes: Page 23, line 19 to page 25, line 2.

Discussion: Page 33, line 14 to page 34, line 23.

Refrences:

1. Babashpour S, Aminzadeh S, Farrokhi N, Karkhane A, Haghbeen K. Characterization of a Chitinase (Chit62) from Serratia marcescens B4A and Its Efficacy as a Bioshield Against Plant Fungal Pathogens. Biochemical Genetics. 2012;50(9):722-35. doi: 10.1007/s10528-012-9515-3.

2. Mosallatpour S, Aminzadeh S, Shamsara M, Hajihosseini R. Novel halo- and thermo-tolerant Cohnella sp. A01 L-glutaminase: heterologous expression and biochemical characterization. Scientific Reports. 2019;9(1):19062. doi: 10.1038/s41598-019-55587-9.

---

## [Decision Letter · Decision Letter 2]

8 Jun 2020

Structural and biochemical characterization of a novel thermophilic Coh01147 protease

PONE-D-19-30576R2

Dear Dr. Aminzadeh,

We’re pleased to inform you that your manuscript has been judged scientifically suitable for publication and will be formally accepted for publication once it meets all outstanding technical requirements.

Kind regards,

Paulo Lee Ho, Ph.D.

Academic Editor

PLOS ONE

Additional Editor Comments (optional):

Reviewers' comments:

Reviewer's Responses to Questions

**Comments to the Author**

1. If the authors have adequately addressed your comments raised in a previous round of review and you feel that this manuscript is now acceptable for publication, you may indicate that here to bypass the “Comments to the Author” section, enter your conflict of interest statement in the “Confidential to Editor” section, and submit your "Accept" recommendation.

Reviewer #3: All comments have been addressed

Reviewer #4: All comments have been addressed

2. Is the manuscript technically sound, and do the data support the conclusions?

Reviewer #3: Yes

Reviewer #4: Yes

3. Has the statistical analysis been performed appropriately and rigorously? 

Reviewer #3: Yes

Reviewer #4: Yes

4. Have the authors made all data underlying the findings in their manuscript fully available?

Reviewer #3: Yes

Reviewer #4: Yes

5. Is the manuscript presented in an intelligible fashion and written in standard English?

Reviewer #3: Yes

Reviewer #4: Yes

6. Review Comments to the Author

Reviewer #3: The authors of the manuscript have addressed all my previous comments, and I do not have any additional comments.

Reviewer #4: This secondly revised version of the manuscript entitled "Structural and biochemical characterization of a novel thermophilic Coh01147 protease", and numbered PONE-D-19-30576R2, can be published in the journal PLOS ONE.

7. PLOS authors have the option to publish the peer review history of their article (what does this mean?). If published, this will include your full peer review and any attached files.

Reviewer #3: No

Reviewer #4: Yes: Prof. Emmanuel M. Papamichael PhD

---

## [Editor Report · Acceptance letter]

12 Jun 2020

PONE-D-19-30576R2 

Structural and biochemical characterization of a novel thermophilic Coh01147 protease 

Dear Dr. Aminzadeh:

I'm pleased to inform you that your manuscript has been deemed suitable for publication in PLOS ONE. Congratulations! Your manuscript is now with our production department. 

Kind regards, 

on behalf of

Dr. Paulo Lee Ho 

Academic Editor

PLOS ONE